# When SNN meets ANN: Error-Free ANN-to-SNN Conversion for Extreme Edge Efficiency

**Gourav Datta**                                                    *gourav.datta@case.edu*
*Department of Electrical, Computer, and Systems Engineering*
*Case Western Reserve University*

**Zeyu Liu**                                                            *liuzeyu@usc.edu*
*Ming Hsieh Department of Electrical and Computer Engineering*
*University of Southern California*

**James Diffenderfer**                                             *diffenderfer2@llnl.gov*
*Lawrence Livermore National Laboratory*

**Bhavya Kailkhura**                                                *kailkhura1@llnl.gov*
*Lawrence Livermore National Laboratory*

**Peter A. Beerel**                                                    *pabeerel@usc.edu*
*Ming Hsieh Department of Electrical and Computer Engineering*
*University of Southern California*

**Reviewed on OpenReview:** *https: // openreview. net/ forum? id= WOwQKguWTO*

## Abstract

Spiking Neural Networks (SNN) are now demonstrating comparable accuracy to convolutional neural networks (CNN), thanks to advanced ANN-to-SNN conversion techniques, all while delivering remarkable energy and latency efficiency when deployed on neuromorphic hardware. However, these conversion techniques incur a large number of time steps, and high spiking activity. In this paper, we propose a novel ANN-to-SNN conversion framework, that incurs an exponentially lower number of time steps compared to that required in the existing conversion approaches. Our framework modifies the standard integrate-and-fire (IF) neuron model used in SNNs with no change in computational complexity and shifts the bias term of each batch normalization (BN) layer in the trained ANN. To reduce spiking activity, we propose training the source ANN with a fine-grained $\ell_1$ regularizer with surrogate gradients that encourages high spike sparsity in the converted SNN. Our proposed framework thus yields lossless SNNs with *low latency*, *low compute energy*, thanks to the low timesteps and high spike sparsity, and *high test accuracy*, for example, 75.12% with only 4 time steps on the ImageNet dataset. Code is available at `https://github.com/godatta/SNN_meets_ANN`.

## 1 Introduction

Spiking Neural Networks (SNNs) (Maass, 1997) have emerged as an attractive spatio-temporal computing paradigm for a wide range of complex computer vision (CV) tasks (Pfeiffer et al., 2018). SNNs compute and communicate via binary spikes that are typically sparse and require only accumulate operations in their convolutional and linear layers, resulting in significant compute efficiency. However, training deep SNNs has been historically challenging, because the spike activation function in standard neuron models in SNNs yields gradients that are zero almost everywhere. While there has been extensive research on backpropagation through time (BPTT) to mitigate this issue (Bellec et al., 2018; Neftci et al., 2019; O'Connor et al., 2018; Wu et al., 2018; Zenke and Ganguli, 2018; Meng et al., 2022a; Xiao et al., 2022), training deep SNNs from scratch is often unable to yield the same accuracies as traditional iso-architecture Artificial Neural Networks (ANN).

ANN-to-SNN conversion, which leverages the advances in state-of-the-art (SOTA) ANN training strategies, has the potential to mitigate this accuracy concern (Sengupta et al., 2019; Rueckauer et al., 2017; Fang et al., 2021). However, since the full-precision ANN activations need to be approximated by binary spikes in the SNN layers, the number of SNN inference time steps required is high. To improve the trade-off between accuracy and time steps, previous research proposed shifting the SNN bias (Deng and Gu, 2021) and initial membrane potential (Bu et al., 2022a; Hao et al., 2023a;b), while leveraging quantization-aware training in the ANN domain (Bu et al., 2022b; Hu et al., 2023a; Schaefer and Joshi, 2020; Sorbaro et al., 2020), inspired by the straight-through estimator method (Bengio et al., 2013). Although this can eliminate the component of the ANN-to-SNN conversion error incurred by the spike-driven binarization, the uneven distribution of the time of arrival of the spikes causes errors, thereby degrading the SNN accuracy. We first uncover that this unevenness error is responsible for the accuracy drop in the converted SNNs in low timesteps. To completely eliminate this unevenness as well as other errors with respect to the quantized ANN, we propose a novel conversion framework that enables exactly identical ANN and SNN activation outputs, while honoring the accumulate-only operation paradigm of SNNs. Our framework: ($i$) encodes both the timing information and binary value of the spikes in the membrane potential with negligible compute overhead, ($ii$) shifts the bias term of the BN layers in the source ANN, and ($iii$) modifies the IF neuron model *with no change in computational complexity* by postponing the neuronal firings and resets after accumulation of the total input current. Our framework yields SNNs with SOTA accuracies among both ANN-to-SNN conversion and BPTT approaches with only $2-4$ time steps.

In summary, we make the following contributions.

- We analyze the key sources of error that ($i$) persist in SOTA ANN-to-SNN conversion approaches, and ($ii$) degrade the SNN accuracy when using low number of time steps.

- We propose a novel ANN-to-SNN conversion framework that exponentially reduces the number of time steps required for SOTA accuracy and eliminates each ANN-to-SNN conversion error. Our resulting SNN can be supported in neuromorphic chips, including Loihi (Davies et al., 2018).

- We significantly increase the compute efficiency of SNNs by incorporating an additional loss term in our training framework, that penalizes the non-zero bits of the intermediate ANN activations, along with the task-specific loss (e.g., cross-entropy for image recognition). Further, we propose a novel surrogate gradient method to optimize this loss.

Our contributions simultaneously provide lower latency, higher energy efficiency compared to existing SNNs, especially those trained using ANN-to-SNN conversion. Additionally, our method surpasses most existing SNN training approaches in terms of performance-efficiency trade-off, as shown in Fig. 1. Note that while we use a modified integrate-and-fire mechanism (postponed resets) inspired by quantized ANN processing, the model operates with binary spikes and retains all the practical attributes of SNNs (sparse spikes, accumulate operations, threshold checks, etc.).

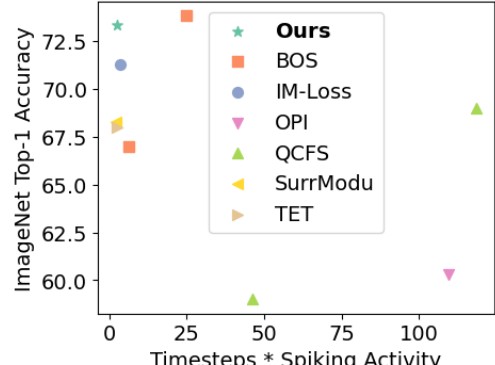

Figure 1: Comparison of the performance-efficiency trade-off between ourconversion & SOTA SNN training methods on ImageNet.

## 2 Related Works

ANN-to-SNN conversion involves estimating the threshold value in each layer by approximating the activation value of ReLU neurons with the firing rate of spiking neurons (Cao et al., 2015; Rueckauer et al., 2017; Diehl et al., 2015; Sengupta et al., 2019; Hu et al., 2018). Some conversion works estimated this threshold using heuristic approaches, such as using the maximum (or close to) ANN preactivation value (Rathi et al., 2020a). Others proposed weight normalization techniques

while setting the threshold to unity (Kim et al., 2019; Sengupta et al., 2019) . While these approaches helped SNNs achieve competitive classification accuracy on the Imagenet dataset, they required hundreds of time steps for SOTA accuracy. Consequently, there has been a plethora of research (Deng and Gu, 2021; Bu et al., 2022b; Hao et al., 2023a;b) that helped reduce the conversion error while also reducing the number of time steps by an order of magnitude. All these works used trainable thresholds in the ReLU activation function in the ANN and reused the same for the SNN threshold. In particular, Deng and Gu (2021); Li et al. (2021a) proposed a shift in the bias term of the convolutional layers to minimize the conversion error, with the assumption that the ANN and SNN input activations are uniformly and identically distributed. Other works include burst spikes (Park et al., 2019; Li and Zeng, 2022), and signed neuron with memory (Wang et al., 2022a). However, they might not adhere to the bio-plausibility of spiking neurons. Some works also proposed modified ReLU activation functions in the source ANN, including StepReLU (Wang et al., 2023a) and SlipReLU (Jiang et al., 2023) to reduce the conversion error. Moreover, there have been works that aim to minimize the deviation error, including Bu et al. (2022b) which proposed to initialize the membrane potential with half of the threshold value; Hao et al. (2023a;b) which adjusts the membrane potential after observing its trend for a few time steps, and Meng et al. (2022b) which proposed threshold tuning and residual block restructuring. Some other works explored error correction methods between ANN and SNNs, often by adapting SNNs through conversion approaches to resemble ANNs more closely (Schaefer and Joshi, 2020; Sorbaro et al., 2020; Hu et al., 2023a). Lastly, some works minimized the conversion error using novel neuron models, such as inverted LIF neuron (Liu et al., 2022) and signed IF neuron (Hu et al., 2023b). Our method builds upon these foundations by focusing specifically on addressing accuracy gaps at very low time steps (e.g., $T \sim 2-4$), while providing substantial computational efficiency.

In contrast to ANN-to-SNN conversion, direct SNN training methods, based on BPTT, aim to resolve the discontinuous and non-differentiable nature of the thresholding-based activation function in the IF model. Most of these methods (Lee et al., 2016; Panda and Roy, 2016; Bellec et al., 2018; Neftci et al., 2019; O'Connor et al., 2018; Wu et al., 2018; 2021; Zenke and Ganguli, 2018; Zenke and Vogels, 2021; Meng et al., 2022a; Xiao et al., 2022; Meng et al., 2023; Guo et al., 2022a) replace the spiking neuron functionality with a differentiable model, that can approximate the real gradients (that are zero almost everywhere) with the surrogate gradients. In particular, Guo et al. (2023a) and Guo et al. (2022b) proposed a regularizing loss and an information maximization loss respectively to adjust the membrane potential distribution in order to reduce the quantization error due to spikes. Some works optimized the BN layer in the SNN to achieve high performance. For example, Duan et al. (2022) proposed temporal effective BN, that rescales the presynaptic inputs with different weights at each time-step; Zheng et al. (2021) proposed threshold-dependent BN; Kim et al. (2020) proposed batch normalization through time that decouples the BN parameters along the temporal dimension; Guo et al. (2023b) used an additional BN layer to normalize the membrane potential. Some works extended direct SNN training to Transformers. For example, Zhou et al. (2023) introduced Spikformer using surrogate gradient learning and spike-based self-attention, demonstrating improved energy efficiency and temporal processing capabilities. Later, Yao et al. (2023) extended this approach with Spike-driven Transformers, where they reformulated attention mechanisms using spike-based computations, later advancing to Spike-driven Transformer V2 (Yao et al., 2024), which introduces meta-spiking architectures optimized for adaptability and next-generation neuromorphic hardware. Lastly, some works have explored the use of $\ell_1$ regularizers in SNN training to improve sparsity (Narduzzi et al., 2022; Ho and Chang, 2021), but to the best of our knowledge, no research has specifically applied this regularization method for ANN-to-SNN conversion.

## 3 Preliminaries

### 3.1 ANN & SNN Neuron Models

For ANNs used in this work, a block $l$ that takes $a_{l-1}$ as input, consists of a convolution (denoted by $f^{conv}$), batchnorm (denoted by $f^{BN}$), and nonlinear activation (denoted by $f^{act}$), as shown below.

$$a^l \quad = f^{act}(f^{BN}(f^{conv}(a^{l-1}))) = f^{act}(z^l) = f^{act}\left(\gamma^l\left(\frac{W^l a^{l-1} - \mu^l}{\sigma^l}\right) + \beta^l\right), \tag{1}$$

where $W^l \in \mathbb{R}^{c_{out} \times c_{in} \times k \times k}$ denotes the convolutional layer weights ($c_{out}$ and $c_{in}$ are the number of output and input channels, respectively, and $k$ is the kernel size), $\mu^l$ and $\sigma^l$ denote the BN running mean and variance, and

$\gamma^l$ and $\beta^l$ denote the learnable scale and bias BN parameters. All BN parameters are vectors with dimensions matching the number of output channels. Inspired by (Bu et al., 2022b), we use quantization-clip-floor-shift (QCFS) as the activation function $f^{act}(\cdot)$ defined as

$$a^l = f^{act}(z^l) = \frac{\lambda^l}{Q}\text{clip}\left(\left\lfloor \frac{z^l Q}{\lambda^l} + \frac{1}{2}\right\rfloor, 0, Q\right),\tag{2}$$

where $a^l \in \mathbb{R}^{N \times c_{\text{out}} \times h \times w}$, $Q$ denotes the number of quantization steps, $\lambda^l$ denotes the scalar trainable QCFS activation output threshold, and $z^l$ denotes the activation input. Note that N is the batch size, and h,w are the spatial dimensions. The *clip* function is represented as

$$\text{clip}(x, 0, \mu) = \begin{cases} 0, & \text{if } x < 0 \\ x, & \text{if } 0 \le x \le \mu \\ \mu, & \text{if } x \ge \mu \end{cases}\tag{3}$$

QCFS can enable ANN-to-SNN conversion with minimal error for arbitrary $T$ and $Q$, where $T$ denotes the total number of SNN time steps.

The spike-driven dynamics of an SNN is typically represented by the IF model where, at each time step denoted as $t$, each neuron integrates the input current $z^l(t)$ from the convolution, followed by BN layer, into its respective state, referred to as membrane potential denoted as $u^l(t)$. The neuron emits a spike if the membrane potential crosses a threshold value, denoted as $\theta^l$. Assuming $s^{l-1}(t)$ and $s^l(t)$ are the spike inputs and outputs respectively, $\mu^l$ and $\sigma^l$ are the BN running mean and variance respectively, and $\gamma^l$ and $\beta^l$ are the learnable scale and bias BN parameters, respectively, the IF model dynamics can be represented as

$$z^l(t) = \left(\gamma^l\left(\frac{W^l s^{l-1}(t)\theta^{l-1} - \mu^l}{\sigma^l}\right) + \beta^l\right), \quad s^l(t) = H(u^l(t-1) + z^l(t) - \theta^l),\tag{4}$$

$$u^l(t) = u^l(t-1) + z^l(t) - s^l(t)\theta^l.\tag{5}$$

where $H(\cdot)$ denotes the heaviside function. Note that $u^l(t)$, $z^l(t)$, and $s^l(t) \in \mathbb{R}^{N \times c_{\text{out}} \times h \times w}$, where $u^l(t)$ and $z^l(t)$ are full-precision tensors while $s^l(t)$ is a binary tensor. Instead of resetting the membrane potential to zero after the spike firing, we use the reset-by-subtraction scheme where the surplus membrane potential over the firing threshold is preserved and propagated to the subsequent time step.

## 3.2 ANN-to-SNN Conversion

The primary goal of ANN-to-SNN conversion is to approximate the SNN spike firing rate with the multi-bit nonlinear activation output of the ANN with the other trainable parameters being copied from the ANN to the SNN. In particular, rearranging Eq. 4 to isolate the expression for $s^l(t)\theta^l$, summing for $t=1$ to $t=T$, and dividing both sides by $T$, we obtain

$$\frac{\sum_{t=1}^T s^l(t)\theta^l}{T} = \frac{\sum_{t=1}^T z^l(t)}{T} + \left(\frac{u^l(0) - u^l(T)}{T}\right).\tag{6}$$

Then, substituting

$$\phi^l(T) = \frac{\sum_{t=1}^T s^l(t)\theta^l}{T}, \text{ and } Z^l(T) = \frac{\sum_{t=1}^T z^l(t)}{T}$$

to denote the average spiking rate and presynaptic potential for the layer $l$ respectively, we obtain

$$\phi^l(T) = Z^l(T) - \left(\frac{u^l(T) - u^l(0)}{T}\right)\tag{7}$$

Note that for a very large $T$, $\phi^l(T)$ can be approximated with $Z^l(T)$. Importantly, the resulting function $\phi^l(T)$ is equivalent to the ANN ReLU activation function, because it outputs zero for negative values of the

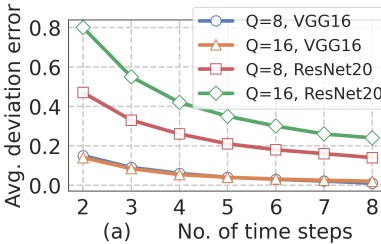 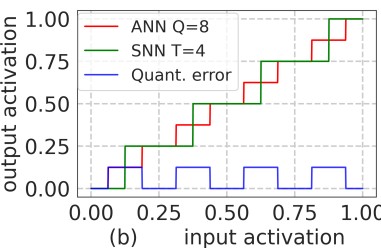 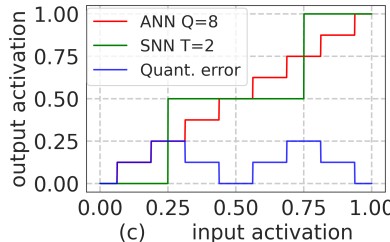

Figure 2: (a) Comparison between the average magnitude of unevenness error for different number of time steps with $Q$=8 and $Q$=16. Comparisons of the SNN and ANN output activations, $\phi^l(T)$ and $a^l$ respectively for (b) Q=8 and T=4, (c) Q=8 and T=2. Reducing the number of time steps from 4 to 2 increases the expected quantization error from $0.0625\lambda^l$ to $0.125\lambda^l$.

input (since the accumulated input current is zero when negative) and directly reflects the positive values of the input current. This analogy is essential in understanding the transition from SNNs to ANNs using spike-based models. However, for the low $T$ in our use-case, the residual term $\left(\frac{u^l(T)-u^l(0)}{T}\right)$ introduces error in the ANN-to-SNN conversion error, which previous works (Hao et al., 2023b;a; Bu et al., 2022b) refer to as *unevenness* error. These works also took into account two other types of conversion errors, namely *quantization* and *clipping* errors. Quantization error occurs due to the discrete nature of $\phi^l(T)$ which has a quantization resolution (QR) of $\frac{\theta^l}{T}$. Clipping error occurs due to the upper bound of $\phi^l(T) = \theta^l$. However, both these errors can be eliminated with the QCFS activation function in the source ANN (see Eq. 2) and setting $\theta^l = \lambda^l$ and $T$=$Q$. This yields the same QR of $\frac{\theta^l}{T}$ and upper bound of $\theta^l$ as the ANN activation.

## 4 Analysis of Conversion Errors

Although we can eliminate the quantization error by setting $T$=$Q$, the error increases as $T$ is decreased significantly from $Q$ for low-latency SNNs[1]. This is because the absolute difference between the ANN activations and SNN average post-synaptic potentials increases as $(Q-T)$ increases as shown in Fig 2(b)-(c). Note that $Q$ cannot be too small, otherwise, the source ANN cannot be trained with high accuracy. To mitigate this concern, we propose to improve the SNN capacity at low $T$ by embedding the information of both the timing and the binary value of spikes in each membrane potential. As shown later in Section 5, this eliminates the *quantization error* at $T$=$\log_2 Q$. This results in an exponential drop in the number of time steps compared to prior works that require $T$=$Q$ (Bu et al., 2022b). As our work already enables a small value of $T$, the drop in SNN performance with further lower $T$<$\log_2 Q$ becomes negligible compared to prior works. Moreover, at low timesteps, the *unevenness error* increases as shown in Fig. 2(a), and even dominates the total error as shown in Fig. 3(Right), which highlights its importance for our use case. Previous works (Hao et al., 2023a;b) attempted to reduce this error by observing and shifting the membrane potential after some number of time steps, which dictates the upper bound of the total latency. Moreover, Hao et al. (2023a) requires iterative potential correction by injecting or eliminating one spike per neuron at a time, which also increases the inference latency. That said, the unevenness error is difficult to overcome with the current IF models. To eliminate the unevenness error, $u^l(T)$ must fall in the range $[0, \theta^l]$ (Bu et al., 2022b). However, this cannot be guaranteed without the prior information of the post-synaptic potentials (up to $T$ time steps). The key reason this cannot be guaranteed is the neuron reset mechanism, which dynamically lowers the post-synaptic potential value based on the input spikes. By shifting all neuron resets to the last time step $T$, and matching the ANN activation and SNN post-synaptic values at each time step, *we can completely eliminate* this *unevenness error*, as shown in Section 5.

---

[1]Note that $T$ cannot always be equal to $Q$ for practical purposes, since we may want multiple SNNs with different number of time steps from a single pre-trained ANN

# 5 Proposed Method

In this section, we propose our ANN-to-SNN conversion framework, which involves training the source ANN using the QCFS activation function (Bu et al., 2022b) and a 1) *bit-wise fine-grained $\ell_1$ regularizer*, followed by 2) *shifting the bias term of the BN layers*, and 3) *modifying the IF model* where the neuron spiking mechanism and reset are pushed after the current accumulation over all the time steps.

## 5.1 ANN-to-SNN Conversion

To enable lossless ANN-to-SNN conversion, the IF layer output should be equal to the bit-wise representation of the output of the corresponding QCFS layer in the $l^{th}$ block, which can be represented as $s^l(t)=a^l_t\ \forall t\in[1,T]$, where $a^l_t$ denotes the $t^{th}$ bit of $a^l$ starting from the most significant bit. This ensures that the cumulative spike train over $T=\log_2 Q$ time steps reconstructs the full quantized activation value of the ANN.

We first show how this is guaranteed for the input block and then for any hidden block $l$ by induction.

**Input Block**: Similar to prior works targeting low-latency SNNs (Bu et al., 2022b;a; Rathi et al., 2020b), we directly use multi-bit inputs that incur multiplications in the first layer, whose overhead is negligible in a deep SNN. Hence, the input to the first IF layer in the SNN (output of the first convolution, followed by BN layer) is identical to the first QCFS layer in the ANN. The first QCFS layer yields the output $a^1$ with $T=\log_2 Q$ bits. The first IF layer also yields identical outputs $s^1(t)=a^1_t$ at the $t^{th}$ time step, with the proposed neuron model as shown later in Eqs. 9 and 10.

**Hidden Block**: To incorporate the information of both the firing time and binary value of the spikes, we multiply the input $s^{l-1}(t)$ of the IF layer (i.e., output of the convolution followed by a BN layer) in the $l^{th}$ block by $2^{(t-1)}$ at the $t^{th}$ time step, which can be easily implemented by a left shifter. Note that the additional compute overhead due to the shifting is negligible as shown later in Section 6.3. The resulting SNN input current in the $l^{th}$ block is computed as $\hat{z}^l(t)=f^{BN}(f^{conv}(2^{t-1}s^{l-1}(t)))$. The input of the corresponding ANN QCFS layer is $f^{BN}(f^{conv}(a^{l-1}))$ where $a^{l-1}$ can be denoted as $\sum_{t=1}^{T}2^{t-1}s^{l-1}(t)$ by induction.

*Condition I*: For lossless conversion, let us first satisfy that the accumulated input current over $T$ time steps is equal to the input of the corresponding QCFS layer in the $l^{th}$ block.

Mathematically, representing the composite function $f^{BN}(f^{conv}(\cdot))$ as $g^{ANN}$ and $g^{SNN}$ for the source ANN and its converted SNN respectively, Condition I can be re-written as

$$\sum_{t=1}^{T}g^{SNN}(k\cdot s^{l-1}(t))=g^{ANN}\left(\sum_{t=1}^{T}k\cdot s^{l-1}(t)\right)\tag{8}$$

where $k=2^{t-1}$. However, this additive property does not hold for any arbitrary source ANN and its converted SNN, due to the BN layer. We satisfy this property by modifying the bias of each BN layer during ANN-to-SNN conversion, as shown in Proposition I below, whose proof is in Appendix A.2.

*Proposition I*: For the $l^{th}$ block in the source ANN, let us denote $W^l$ as the weights of the convolutional layer, and $\mu^l$, $\sigma^l$, $\gamma^l$, and $\beta^l$ as the trainable parameters of the BN layer. Let us denote the same parameters of the converted SNN for as $W^l_c$, $\mu^l_c$, $\sigma^l_c$, $\gamma^l_c$, and $\beta^l_c$. Then, Eq. 8 holds true if $W^l_c=W^l$, $\mu^l_c=\mu^l$, $\sigma^l_c=\sigma^l$, $\gamma^l_c=\gamma^l$, and $\beta^l_c=\frac{\beta^l}{T}+(1-\frac{1}{T})\frac{\gamma^l\mu^l}{\sigma^l}$.

*Proposition II*: If Condition I (Eq. 8) is satisfied and the post-synaptic potential accumulation, neuron firing, and reset model adhere to Eqs. 9 and 10 below, the lossless conversion objective i.e., $s^l(t)=a^l_t\ \forall t\in[1,T]$ is satisfied for any hidden block $l$.

$$\hat{z}^l(t)=\left(\gamma^l_c\left(\frac{2^{t-1}W^l_c s^{l-1}(t)\theta^{l-1}-\mu^l_c}{\sigma^l_c}\right)+\beta^l_c\right),\tag{9}$$

$$u^l(1)=\sum_{t=1}^{T}\hat{z}^l(t),\quad s^l(t)=H\left(u^l(t)-\frac{\theta^l}{2^t}\right),\quad u^l(t+1)=u^l(t)-s^l(t)\frac{\theta^l}{2^t}.\tag{10}$$

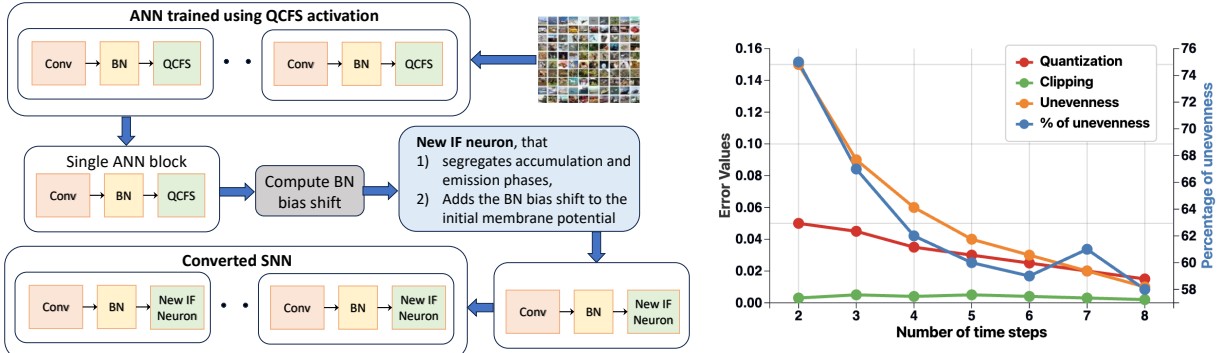

Figure 3: (Left) Proposed ANN-to-SNN conversion framework, encompassing i) training of the source ANN using the QCFS activation function, ii) computing the shift of the bias term of the BN layers, and copying the other trainable parameters and iii) modification of the IF neuron. (Right) Comparison of the average magnitude of quantization, clipping, and unevenness errors between the ANN and SNN.

The proof of Proposition II is shown in Appendix A.2. Our conversion framework is illustrated in Fig. 3(Left). Note that our neuron model postpones the firing and reset mechanism until after the input current is accumulated from the incoming spikes emitted over all the T time steps in the previous layer. Hence, our model does not change the computational complexity of the traditional IF model. Moreover, our neuron model can be supported in programmable neuromorphic chips, that implements current accumulation, threshold comparison, and potential reset independently in a modular fashion. Since our model needs to acquire $\hat{z}^l(T)$, before transmitting the spikes at any time step to the subsequent layer, it requires layer-by-layer propagation, as used in advanced conversion works (Hao et al., 2023a;b). However, *this does not prohibit the asynchronous computations that can be accelerated by an asynchronous accelerator such as Loihi. In particular, spikes are transmitted to the next layer as soon as they are computed. Moreover, our implemented framework adheres to this scheme and thus our reported accuracies are consistent with the asynchronous implementation.* The only constraint the layer-by-layer propagation incurs is that all time steps of the previous layer must be computed before the spikes of the first time step of the next layer can be computed. However, this constraint does not impose any penalty, as layer-by-layer propagation is superior compared to its alternative step-by-step propagation in terms of system efficiency as shown in Appendix A.3.

While our approach of separating the aggregation and emission phases is similar to Liu et al. (2022), there are notable differences that result in improved SNN accuracy, particularly at low time steps. Firstly, our method embeds both the timing and binary value of spikes within the accumulated input current (as indicated by the term $2^{t-1}$ in Eq. 9). Secondly, we provide a mathematical proof demonstrating that our proposed neuron model completely eliminates the conversion error, in contrast, Liu et al. (2022) empirically shows that their inverted LIF model only reduces (not eliminate) the conversion error.

## 5.2 Activation Sparsity

Although our proposed framework can significantly reduce $T$ while eliminating the conversion error, the spiking activity does not reduce proportionally. Note that spiking activity is defined as the average number of spikes per neuron in the entire SNN over all the time steps i.e., the total inferencing window. In fact, we can see from Fig. 7(a) that the spiking activity of a VGG-16 based SNN evaluated on CIFAR10 drops only ~3% (36.2% to 33.0%) when $T$ decreases from 8 to 4. We hypothesize this is because the SNN tries to pack a similar number of spikes within the few time steps available. To mitigate this concern, we propose a fine-grained regularization method that encourages more zeros in the bit-wise representation of the source ANN. As our approach enforces similarity between the SNN spiking and ANN bit-wise output, this encourages more spike sparsity under low $T$, which in turn, decreases the compute complexity of the SNN when deployed on neuromorphic hardware. The training loss function ($L_{total}$) of our proposed approach is shown below in Eq. 11, where $a_t^{i,l}$ denotes the $t^{th}$ bit of the $i^{th}$ activation value in layer $l$, $L_{CE}$ denotes the cross-entropy loss calculated on the softmax output of the last layer $L$, $L_{SP}$ denotes the proposed $\ell_1$ regularizer loss, and $\lambda$ is

the regularization coefficient.

$$L_{total}=L_{CE} + \lambda L_{SP}=L_{CE}+\lambda \sum_{l=1}^{L-1}\sum_{t=1}^{T}\sum_{i=1}^{N}a_t^{i,l}. \tag{11}$$

Note that we only accumulate (and do not spike) the post-synaptic potential in the last layer $L$, and hence, we do not incorporate the loss due to $a_t^{i,l}$ for $l=L$. Since $a_t^{i,l}\in\{0,1\}$, its gradients are either zero or undefined, and so, we cannot directly optimize $L_{SP}$ using backpropagation. To mitigate this issue, inspired by the straight-through estimator (Bengio et al., 2013), we propose a form of surrogate gradient descent as shown below, where $a^{i,l}$ denotes the $t$-bit activation of neuron $i$ in layer $l$:

$$\frac{\partial L_{SP}}{\partial a^{i,l}}=\lambda \sum_{l=1}^{L}\sum_{i=1}^{N}\sum_{t=1}^{T}\frac{\partial a_t^{i,l}}{\partial a^{i,l}}, \quad \frac{\partial a_t^{i,l}}{\partial a^{i,l}}=\begin{cases}1, & \text{if } 0 < a^{i,l} < \lambda^l \\ 0, & \text{otherwise}\end{cases} \tag{12}$$

## 6 Experimental Results

In this section, we demonstrate the efficacy of our framework on image recognition tasks with CIFAR-10 (Lecun et al., 1998), CIFAR100 (Krizhevsky, 2009), and ImageNet datasets (Deng et al., 2009). Similar to prior works, we evaluate our framework on VGG-16 (Simonyan and Zisserman, 2014), ResNet18 (He et al., 2016), ResNet20, and ResNet34 architectures for the source ANNs. To the best of our knowledge, we are the first to yield low latency SNNs based on the MobileNetV2 (Sandler et al., 2018) architecture. We compare our method with the SOTA ANN-to-SNN conversion methods including Rate Norm Layer (RNL) (Ding et al., 2021), Signed Neuron with Memory (SNM) (Wang et al., 2022a), radix encoded SNN (radix-SNN) (Wang et al., 2022b), SNN Conversion with Advanced Pipeline (SNNC-AP) (Li et al., 2021a), Optimized Potential Initialization (OPI) (Bu et al., 2022a), QCFS (Bu et al., 2022b), Bridging Offset Spikes (BOS) (Hao et al., 2023a), Residual Membrane Potential (SRP) (Hao et al., 2023b) and direct training methods including Dual Phase (Wang et al., 2023b), Diet-SNN (Rathi et al., 2020b), Information loss minimization (IM-Loss) (Guo et al., 2022b), Differentiable Spike Representation (DSR) (Li et al., 2021b), Temporal Efficient Training (Deng et al., 2022), parametric leaky-integrate-and-fire (PLIF) (Fang et al., 2021), RecDis-SNN (Guo et al., 2022a), Membrane Potential Reset (MPR) (Guo et al., 2022c), Temporal Effective Batch Normalization (TEBN) (Duan et al., 2022), and Surrogate Module Learning (SML) (Deng et al., 2023). More details about the proposed conversion algorithm and training configurations are in Appendix A.1.

### 6.1 Efficacy of Proposed Method

To verify the efficacy of our proposed method, we compare the accuracies obtained by our source ANN and the converted SNN on CIFAR datasets. As shown in Fig. 4, for both VGG and ResNet architectures, the accuracies obtained by our source ANN and converted SNN are identical for $T=log_2Q$. This is expected since we ensure that both the ANN and SNN produce the same activation outputs with the shift of the bias term of each BN layer. Hence, unlike previous works, there is no layer-wise error that gets accumulated and transmitted to the output

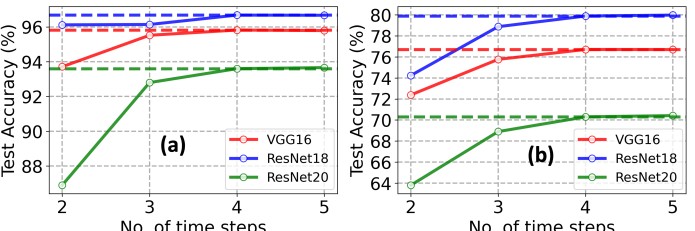

Figure 4: Comparison of the test accuracy of our conversion method for different time steps with $Q = 16$ on (a) CIFAR10 and (b) CIFAR100 datasets. For $T=log_2Q=4$, the ANN & SNN test accuracies are identical. The source ANN accuracies are shown in dotted lines.

layer. However, the SNN test accuracy starts reducing for lower $T$, which is due to the difference between the ANN and SNN activation outputs, but is still higher than existing works at the same $T$ as shown below.

| Architecture | Method | ANN | $T$=2 | $T$=4 | $T$=6 | $T$=8 | $T$=16 | $T$=32 |
|---|---|---|---|---|---|---|---|---|
| VGG16 | RNL | 92.82% | - | - | - | - | 57.90% | 85.40% |
| | SNNC-AP | 95.72% | - | - | - | - | - | 93.71% |
| | OPI | 94.57% | - | - | - | 90.96% | 93.38% | 94.20% |
| | BOS* | 95.51% | - | - | 95.36% | 95.46% | 95.54% | 95.61% |
| | Radix-SNN | - | - | 93.84% | 94.82% | - | - | - |
| | QCFS | 95.52% | 91.18% | 93.96% | 94.70% | 94.95% | 95.40% | 95.54% |
| | **Ours** | 95.82% | 94.21% | 95.82% | 95.79% | 95.82% | 95.84% | 95.81% |
| ResNet18 | OPI | 96.04% | - | - | - | 66.24% | 87.22% | 91.88% |
| | BOS* | 95.64% | - | - | 95.25% | 95.45% | 95.68% | 95.68% |
| | Radix-SNN | - | - | 94.43% | 95.26% | - | - | - |
| | QCFS | 95.64% | 91.75% | 93.83% | 94.79% | 95.04% | 95.56% | 95.67% |
| | **Ours** | 96.68% | 96.12% | 96.68% | 96.65% | 96.67% | 96.73% | 96.70% |
| ResNet20 | OPI | 92.74% | - | - | - | 66.24% | 87.22% | 91.88% |
| | BOS* | 91.77% | - | - | 89.88% | 91.26% | 92.15% | 92.18% |
| | QCFS | 91.77% | 73.20% | 83.75% | 83.79% | 89.55% | 91.62% | 92.24% |
| | **Ours** | 93.60% | 86.9% | 93.60% | 93.57% | 93.66% | 93.75% | 93.82% |

Table 1: Comparison of our proposed method to existing ANN-to-SNN conversion approaches on CIFAR10. $Q = 16$ for all architectures, $\lambda$=1$e$−8. *BOS incurs at least 4 additional time steps to initialize the membrane potential, so their results are reported from $T$>4.

| Architecture | Method | ANN | $T$=2 | $T$=4 | $T$=6 | $T$=8 | $T$=16 | $T$=32 |
|---|---|---|---|---|---|---|---|---|
| ResNet34 | SNM | 73.18% | - | - | - | - | - | 64.78% |
| | SNNC-AP | 75.36% | - | - | - | - | - | 63.64% |
| | OPI | 93.63% | - | - | - | - | - | 60.30% |
| | BOS* | 74.22% | - | - | 67.12% | 68.86% | 74.17% | 73.95% |
| | SRP* | 74.32% | - | - | - | 57.22% | 67.62% | 68.18% |
| | Radix-SNN | - | - | 72.52% | 73.45% | 73.65% | - | - |
| | QCFS | 74.32% | - | - | - | 35.06% | 59.35% | 69.37% |
| | **Ours** | 75.12% | 54.27% | 75.12% | 75.00% | 75.02% | 75.10% | 75.14% |
| MobileNetV2 | SNNC-AP | 73.40% | - | - | - | - | - | 37.43% |
| | QCFS | 69.02% | 0.20% | 0.26% | 0.53% | 1.12% | 21.74% | 58.45% |
| | **Ours** | 69.02% | 22.62% | 68.81% | 68.89% | 68.98% | 69.02% | 69.01% |

Table 2: Comparison of our proposed method to existing conversion methods on ImageNet. $Q$=16 for both ResNet34 and MobileNetV2, and $\lambda$=5$e$−10. *BOS and SRP incurs at least 4 and 8 additional time steps to initialize the potential, so their results are reported from $T$>4 and $T$>8 respectively.

## 6.2 Comparison with SOTA

We compare our proposed framework with the SOTA ANN-to-SNN conversion approaches on CIFAR10 and ImageNet in Table 1 and 2 respectively. For a low number of time steps, especially $T\leq4$, the test accuracy of the SNNs trained with our method surpasses all the existing methods. Note that our accuracy slight drops at T=6 likely because the source QCFS ANN being trained with Q=16 (equivalent to 4-bit precision), leading to optimal performance at T=4. However, this drop ($\sim$0.1%) is within expected noise levels. Our SNNs can also outperform some of the recently proposed SNNs that incur even higher number of time steps. For example, QCFS reported a test accuracy of 94.95% at $T$=8; our method can surpass that accuracy (yield 95.82%) at $T$=4. Note that Hao et al. (2023b;a) requires additional time steps to capture the temporal trend of the membrane potential. The authors reported 4 extra time steps for the accuracy numbers that are shown in Table 1. As a result, they require at least 5 time steps during inference, and their reported accuracies

are lower compared to our SNNs at iso-time-step across different architectures and datasets. Moreover, our approach results in >2% increase in test accuracy on both CIFAR10 and ImageNet compared to radix encoding (Wang et al., 2022b), that proposed a shifting method similar to our left-shift approach, for low time steps (<4). This demonstrates the efficacy of our BN bias shift and neuron model. Moreover, as shown in Table 3, our low-latency accuracies are also higher compared to other SOTA yet memory-expensive SNN training techniques, such as BPTT and hybrid training, at iso-time-step. Lastly, compared to these, our conversion approach leverages standard ANN training with QCFS activation and requires changing only one parameter of each BN layer, that is not repeated across time steps, before the SNN inference process.

### 6.3 Energy Efficiency

Our modified IF model incurs the same number of membrane potential update, neuron firing, and reset, compared to the traditional IF model with identical spike sparsity. The only additional overhead is the left shift operation that is performed on each convolutional layer output in each time step. As shown in Table 6 in Appendix A.4, a left shift operation consumes similar energy as an addition operation with identical bit-precision. However, the total number of left shift operations is significantly lower than the number of addition operations incurred in an SNN for the spiking convolution operation. Intuitively, this is because the computational complexity of the spiking convolution operation and the left shift operation are $\mathcal{O}(sk^2 c_{in} c_{out} HW)$ and $\mathcal{O}(c_{out} HW)$ respectively, where $s$ denotes the sparsity. Note that $k$ denotes the kernel size, $c_{in}$ and $c_{out}$ denote the number of input and output channels respectively, and $H$ and $W$ denote the spatial dimensions of the activation map. Even with a sparsity of 90%, for $c_{in}$=512 and $k$=3, in ResNet18, we have $\frac{sk^2 c_{in} c_{out} HW}{c_{out} HW}$=406.8. Hence, as shown in Fig. 5(a), the left shifts incur negligible overhead in the total compute energy, which is the the energy incurred by the floating point operations, specifically accumulate operations in SNNs (Rathi and Roy, 2023; Datta et al., 2024), across both VGG and ResNet architectures. Moreover, left shifts can also be supported in programmable neuromorphic chips.

Our SNNs enjoy superior energy efficiency compared to existing SNNs with a higher number of time steps, as our method effectively reduces the required time steps while preserving accuracy. By maintaining the same computational complexity as a quantized ANN and incorporating modifications to the IF model, our approach enables a more energy-efficient SNN. Moreover, our fine-grained regularizer significantly reduces the spiking activity of the network. As shown in Fig. 5(b)-(c), with VGG16, we can obtain a 1.64× reduction for CIFAR10 and 2.40× reduction

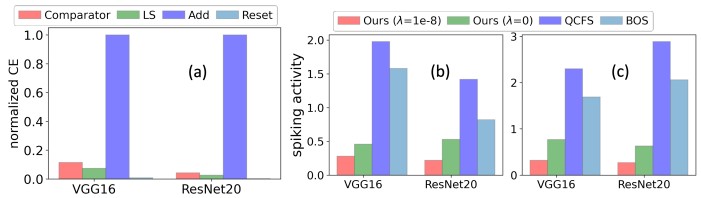

Figure 5: (a) Comparison of the compute energy of each SNN operation with $\lambda$=1$e$−8 on CIFAR10. Comparison of the spiking activites of the SNNs obtained via our and SOTA conversion methods on (b) CIFAR10 and (c) CIFAR100 with VGG16 and ResNet20. In (a), LS denotes the left shift operation, and CE denotes compute energy.

for CIFAR100. For ResNet-18 on CIFAR10 and ResNet-34 on CIFAR100, the reduction factors are 2.41× and 2.33× respectively. Compared to SOTA conversion approaches (Bu et al., 2022b; Hao et al., 2023a), we obtain 3.73−10.70× reduction in spiking activity. This reduced spiking activity linearly reduces the compute energy. Additionally, our low-latency SNNs significantly reduce the memory access cost, which is dominated by the successive *read* and *write* operations of the membrane potentials in each time step. Thus, our proposed low-latency conversion framework, coupled with high spike sparsity, can significantly reduce the combined system energy. Detailed energy comparisons with ANNs and additional analysis are in Appendix A.4.

### 6.4 Ablation study of Neuron Model

We conduct ablation studies of our proposed encoding and conversion framework using the traditional IF model. As shown in Table 4, the SNN accuracy drops compared to the ANN counterpart, and the degradation is severe for low (2-4) time steps. This is due to the deviation error that appears with the normal IF model, and increases significantly at low time steps, dominating the total error. These results validate our hypothesis

| Dataset | Method | Approach | Architecture | Accuracy | Time Steps |
|---|---|---|---|---|---|
| CIFAR10 | Dual-Phase | Hybrid | ResNet18 | 93.27 | 4 |
| | IM-Loss | BPTT | ResNet19 | 95.40 | |
| | MPR | BPTT | ResNet19 | 96.27 | |
| | TET | BPTT | ResNet19 | 94.44 | |
| | RecDis-SNN | BPTT | ResNet19 | 95.53 | |
| | TEBN | BPTT | ResNet19 | 95.58 | |
| | SurrModu | BPTT | ResNet19 | 96.04 | |
| | **Ours** | ANN-to-SNN | ResNet18 | **96.68** | |
| ImageNet | Dspike | Supervised learning | VGG16 | 71.24 | 5 |
| | Diet-SNN | Hybrid | VGG16 | 69.00 | 5 |
| | SEW ResNet | BPTT | ResNet34 | 67.04 | 4 |
| | IM-Loss | BPTT | VGG16 | 70.65 | 5 |
| | RMP-Loss | BPTT | ResNet34 | 65.27 | 4 |
| | SurrModu | BPTT | ResNet34 | 68.25 | 4 |
| | SDT V2 | BPTT | Meta-Spikeformer | 80.00 | 4 |
| | Spikformer V2 | BPTT | Spikformer V2-8-512 | **80.38** | 4 |
| | **Ours** | ANN-to-SNN | ResNet34 | 75.12 | 4 |

Table 3: Comparison of our method with SOTA BPTT and hybrid training approaches.

| Architecture | Left shift | BN bias shift | Modified IF | $T=2$ | $T=4$ | $T=6$ | $T=8$ | $T=16$ |
|---|---|---|---|---|---|---|---|---|
| VGG16 | × | × | × | 91.08% | 93.82% | 94.68% | 94.90% | 95.33% |
| | × | × | ✓ | 92.42% | 94.80% | 95.17% | 95.28% | 95.21% |
| | ✓ | × | × | 93.03% | 95.12% | 95.24% | 95.18% | 95.21% |
| | ✓ | ✓ | × | 93.33% | 95.23% | 95.45% | 95.45% | 95.32% |
| | ✓ | ✓ | ✓ | 94.21% | 95.82% | 95.79% | 95.82% | 95.84% |
| ResNet20 | × | × | × | 71.42% | 83.91% | 84.12% | 88.72% | 92.64% |
| | × | × | ✓ | 76.21% | 90.18% | 91.92% | 92.49% | 92.62% |
| | ✓ | × | × | 76.10% | 91.22% | 91.43% | 92.40% | 92.62% |
| | ✓ | ✓ | × | 79.86% | 91.81% | 92.07% | 93.24% | 93.48% |
| | ✓ | ✓ | ✓ | 86.92% | 93.60% | 93.57% | 93.66% | 93.75% |

Table 4: Ablation study of the different components of our proposed method on CIFAR10 with VGG16 and ResNet20.

presented in Section 3. Additionally, when we use the normal IF model, the encoding and bias shift of the BN layers still yield noticeable accuracy increase compared to the QCFS training method that our work is based on, especially for 2-4 time steps. For hardware that can only support the standard IF model, our conversion framework employing this model yields superior accuracy compared to most of the existing SNN works, as shown in Table 1.

### 6.5 Comparison with quantized ANN

While SNNs were originally proposed to mimic the neural mechanism of humans, SNNs can also yield high energy efficiency arising from the spike sparsity and accumulate-only operations, while maintaining state-of-the-art accuracy. Our method enhances the energy efficiency compared to existing SNNs by drawing inspiration from activation quantized ANNs and proposing a new neuron model and batch norm (BN) bias modification strategy, that ensures the ANN and average SNN outputs are identical at each layer. While this implies some degree of similarity with quantized ANNs, marrying the efficiency benefits from the quantization in ANNs and sparsity in SNNs helps enable low-power and low-latency neural networks, particularly given the rise of neuromorphic chips.

| Architecture | T=Q | SNN Acc. (%) | QANN Acc. (%) |
|---|---|---|---|
| | 2 | 94.21 | 94.73 |
| VGG16 | 3 | 95.30 | 95.37 |
| | 4 | 95.82 | 96.02 |
| | 2 | 86.90 | 86.73 |
| ResNet20 | 3 | 90.77 | 91.22 |
| | 4 | 93.60 | 94.06 |

Table 5: Comparison of accuracy of the SNNs obtained via our conversion framework with quantized ANNs (QANN) on CIFAR10.

As shown in Table 5, with VGG16 and ResNet20 on CIFAR10, our SNNs incur only a marginal reduction of test accuracy compared to quantized ANNs. This reduction is due to our fine-grained $\ell_1$ regularizer that trades accuracy for spiking activity. Note that for a fair comparison, we use $T = Q$, where T is the total number of SNN time steps, and Q is the activation bit-width of the ANN. While our SNNs incur a slight drop in accuracy, they are significantly more energy efficient than quantized ANNs. First, quantized ANN accelerators do not typically leverage activation sparsity that avoid computation when any of the bits in the activation are zero. Secondly, they require quantized multiply-and-accumulate (MAC) operations, which incur significantly more energy compared to accumulate (AC) operations required by SNNs. For example, a 4-bit integer MAC operation incurs $2.3\times$ higher compute energy compared to a 4-bit integer AC operation in 45 nm CMOS technology, as observed in our in-house FPGA simulations. Thirdly, our SNNs provide additional spike sparsity (on top of the natural spike sparsity) due to our fine-grained $\ell_1$ regularizer, which further increases the energy-efficiency. As a result, our SNNs incur $\sim 5.1\times$ lower compute energy for $T = Q = 4$ as shown in Table 11 in Appendix, when averaged over VGG and ResNet architectures, on CIFAR10 and ImageNet.

## 7 Conclusion

In this paper, we first uncover the key sources of error in ANN-to-SNN conversion that have not been completely eliminated in existing works. We propose a novel conversion framework, that introduces a modified IF neuron model and shifts the bias term of each BN layer of the source ANN, before the SNN inference. Our neuron model has identical compute and memory complexity compared to the traditional IF neuron model. Our framework completely eliminates all sources of conversion errors when we use the same number of time steps as the bit precision of the source ANN. We also propose a fine-grained $\ell_1$ regularizer during the source ANN training that minimizes the number of spikes in the converted SNN. This significantly increases the compute efficiency, while the ultra-low latency increases the memory efficiency of our SNNs. To the best of our knowledge, our work is the first to achieve ultra-low latency and compute energy, while still achieving the SOTA test accuracy on complex image recognition tasks with SNNs.

## 8 Acknowledgements

Gourav Datta, Zeyu Liu, and Peter A. Beerel gratefully acknowledge a gift funding from Intel Labs. James Diffenderfer and Bhavya Kailkhura conducted this research under the auspices of the U.S. Department of Energy by the Lawrence Livermore National Laboratory under Contract No. DE-AC52-07NA27344 and was supported by the LLNL-LDRD Program under Project No. 22-DR-009 (LLNL-JRNL-857835).

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

# A  Appendix

## A.1  Network Configurations and Hyperparameters

We train our source ANNs with average-pooling layers instead of max-pooling as used in prior conversion works (Hao et al., 2023a; Bu et al., 2022b). We also replace the ReLU activation function in the ANN with

QCFS function as shown in Eq. 2, copy the weights from the source ANN to the target SNN and set the QCFS activation threshold $\lambda^l$ equal to the SNN threshold $\theta^l$. Note that $\lambda^l$ is a scalar term for the entire layer to minimize the compute associated with the left-shift of the threshold in the SNN. We set the number of quantization steps $Q$ to 16 for all networks on all datasets.

We leverage the Stochastic Gradient Descent optimizer (Bottou, 2012) with a momentum value of 0.9. We use an initial learning rate of 0.02 for CIFAR-10 and CIFAR-100, and 0.1 for ImageNet, with a cosine decay scheduler (Loshchilov and Hutter, 2017) to lower the learning rate. For CIFAR datasets, we set the value of weight decay to $5 \times 10^{-4}$, while for ImageNet, it is set to $1 \times 10^{-4}$. Additionally, we leverage advanced input augmentation techniques to boost the performance of our source ANN models (DeVries and Taylor, 2017; Cubuk et al., 2019), which can eventually improve the performance of our SNNs. The models for CIFAR datasets are trained for 600 epochs, while those for ImageNet are trained for 300 epochs. All experiments are performed on an NVIDIA V100 GPU with 16 GB memory.

## A.2  Proof of Propositions & Statements

*Proposition-I*: For the $l^{th}$ block in the source ANN, let us denote $W^l$ as the weights of the $l^{th}$ hidden convolutional layer, and $\mu^l$, $\sigma^l$, $\gamma^l$, and $\beta^l$ as the trainable parameters of the BN layer. Let us denote the same parameters of the converted SNN for as $W_c^l$, $\mu_c^l$, $\sigma_c^l$, $\gamma_c^l$, and $\beta_c^l$. Then, Eq. 8 holds true if $W_c^l = W^l$, $\mu_c^l = \mu^l$, $\sigma_c^l = \sigma^l$, $\gamma_c^l = \gamma^l$, and $\beta_c^l = \frac{\beta^l}{T} + (1 - \frac{1}{T})\frac{\gamma^l \mu^l}{\beta^l}$.

*Proof*: Substituting the value of $g^{SNN}$ for the SNN in the left-hand side (LHS) which is equal to the accumulated input current over $T$ time steps, $\sum_{t=1}^{T} \hat{z}_l$, and $g^{ANN}$ in the right-hand side (RHS) of Equation 8, we obtain

$$\sum_{t=1}^{T}\left(\gamma_c^l\left(\frac{2^{t-1}W_c^l s^{l-1}(t)\theta^{l-1} - \mu_c^l}{\sigma_c^l}\right) + \beta_c^l\right) = \left(\gamma^l\left(\frac{\sum_{t=1}^{T}(2^{t-1}W^l s^{l-1}(t)\theta^{l-1}) - \mu^l}{\sigma^l}\right) + \beta^l\right)$$

$$\implies \frac{\gamma_c^l W_c^l \theta^{l-1}}{\sigma_c^l}\sum_{t=1}^{T} 2^{t-1}s^{l-1}(t) + T(\beta_c^l - \frac{\mu_c^l \gamma_c^l}{\sigma_c^l}) = \frac{\gamma^l W^l \theta^{l-1}}{\sigma^l}\sum_{t=1}^{T} 2^{t-1}s^{l-1}(t) + (\beta^l - \frac{\mu^l \gamma^l}{\sigma^l})$$

If we assert $\gamma_c^l = \gamma^l$, $W_c^l = W^l$, $\sigma_c^l = \sigma^l$, the first terms of both LHS and RHS are equal. Substituting $\gamma_c^l = \gamma^l$, $W_c^l = W^l$, and $\sigma_c^l = \sigma^l$ with this assertion, LHS=RHS if their second terms are equal, i.e,

$$T(\beta_c^l - \frac{\mu^l \gamma^l}{\sigma^l}) = (\beta^l - \frac{\mu^l \gamma^l}{\sigma^l}) \implies T\beta_c^l = \beta^l + (T-1)\frac{\mu^l \gamma^l}{\sigma^l} \implies \beta_c^l = \frac{\beta^l}{T} + (1 - \frac{1}{T})\frac{\mu^l \gamma^l}{\sigma^l}$$

*Proposition-II*: If Condition I (Eq. 8) is satisfied and the post-synaptic potential accumulation, neuron firing, and reset model adhere to Eqs. 9 and 10, the lossless conversion objective i.e., $s^l(t) = a_t^l \; \forall t \in [1, T]$ is satisfied for any hidden block $l$.

In Eqs. 9 and 10, $u^l(1) = \sum_{t=1}^{T} \hat{z}^l(t)$ is the original LHS of Eq. 8. Given that Eq. 8 is satisfied due to Proposition-I, we can write $u^l(1) = h^l$, where $h^l$ is the input to the QCFS activation function of the $l^{th}$ block of the ANN. The output of the QCFS function is denoted as $a^l = f^{act}(h^l)$, whose $t^{th}$ bit starting from the most significant bit (MSB) is represented as $a_t^l$. We can check if $a_t^l$ is zero or one, iteratively starting from the MSB, using a binary decision tree approach where we progressively discard one-half of the search range for the subsequent bit checking. With the maximum value of $h^l$ being $\lambda^l$, and $\lambda^l = \theta^l$ (see Section 3.2), $a_1^l = H(h^l - \frac{\theta^l}{2}) = H(u^l(1) - \frac{\theta^l}{2}) = s^l(1)$. To compute $a_2^l$, we can lower $h^l$ by half of the previous range, by first updating $h^l$ as $h^l = h^l - a_1^l\frac{\theta^l}{2}$, and then calculating $a_2^l = H(h^l - \frac{\theta^l}{4}) = H(u^l(2) - \frac{\theta^l}{4})$ which is equal to $s^l(2)$. Similarly, updating $h^l$ to calculate the $t^{th}$ bit $\forall \, t \in [2, T]$ as $h^l = h^l - \frac{\theta^l}{2^{t-1}}$ and then evaluating $a_t^l$ as $a_t^l = H(h^l - \frac{\theta^l}{2^t})$, we obtain $a_t^l = s^l(t), \; \forall t \in [1, T]$.

## A.3  Efficacy of layer-by-Layer Propagation

### A.3.1  Spatial Complexity

During the SNN inference, the layer-by-layer propagation scheme incurs significantly lower spatial complexity compared to its alternative step-by-step propagation. This is because in step-by-step inference, the computations are localized at a single time step for all the layers, and to process a subsequent time step, all the data, including the outputs and hidden states of all layers at the previous time step, can be discarded. Thus, the

| Operation | Bit Precision | Energy (pJ) |
|---|---|---|
| Mult. | 32 | 3.1 |
| | 8 | 0.2 |
| Add. | 32 | 0.1 |
| | 8 | 0.03 |
| Left Shift | 32 | 0.13 |
| | 8 | 0.024 |
| Comparator | 32 | 0.08 |
| | 8 | 0.03 |

Table 6: Comparison of the energy consumed by the different operations in our proposed IF neuron model, and multiplication required in ANNs, on an ASIC (45 nm CMOS technology). The energy values for multiply operations are sourced from Horowitz (2014); Gholami et al. (2021), while those for the shift operation are from You et al. (2020); Sekikawa and Yashima (2023). The comparator energy numbers are obtained from our in-house circuit simulations, all within 45nm CMOS technology.

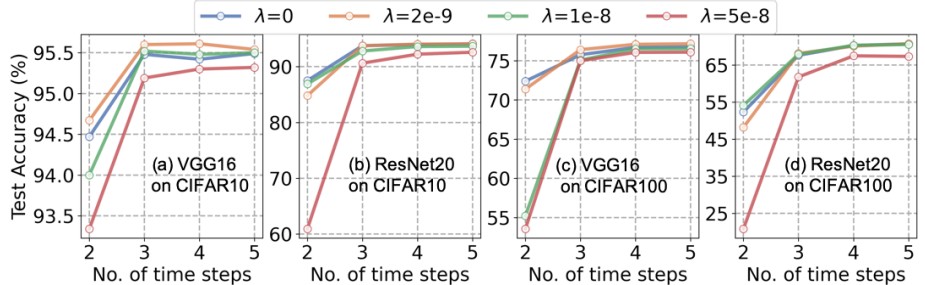

Figure 6: Comparison of the test accuracy of our conversion method for different values of the regularization coefficient $\lambda$.

spatial inference complexity of the step-by-step propagation is $O(N \cdot L)$, which is not proportional to $T$. In contrast, for layer-by-layer propagation, the computations are localized in a single layer, and to process a subsequent layer, all the data of the previous layers can be discarded. Thus, the spatial inference complexity of the layer-by-layer propagation scheme is $O(N \cdot T)$. Since $T << L$ for deep and ultra low-latency SNNs, the layer-by-layer propagation scheme has lower spatial complexity compared to the step-by-step propagation.

### A.3.2 Latency Complexity

When operating with step-by-step propagation scheme, let us assume that the $l^{th}$ layer requires $t_{step}(l)$ to process the input $s^{l-1}(t)$ and yield the output $s^l(t)$. Then, the latency between the input $X$ and the output $s^L(T)$ is $D_{step} = T \sum_{l=1}^{L} t_{step}(l)$.

With layer-by-layer propagation, let us assume that the delay in processing the layer $l$ i.e., outputting the spike outputs for all the time steps $(s^l(t) \ \forall t \in [1, T])$ from the instant the first spike input $s^{l-1}(1)$ is received, is $t_{layer}(l)$. Then, the total latency between the input $X$ and the output $s^L(T)$ is $D_{layer} = \sum_{l=1}^{L} t_{layer}(l)$.

Although each SNN layer is stateful, the computation across the different time steps can be fused into a large CUDA kernel in GPUs when operating with the layer-by-layer propagation scheme (Fang et al., 2023). Even on neuromorphic chips such as Loihi (Davies et al., 2018), there is parallel processing capability. All these imply that $t_{layer}(l) < T \cdot t_{step}(l)$ for any layer $l$. This further implies that $D_{layer} = \sum_{l=1}^{L} t_{layer}(l) < \sum_{l=1}^{L} T \cdot t_{step}(l) < D_{step}$.

For a concrete example, consider a network with $L = 34$ layers (e.g., ResNet-34) running at $T = 4$ time steps. Suppose each layer's per-step compute time $t_{step}(l) \approx 1$ unit (normalized). Then step-by-step latency would be $D_{step} = 4 \times \sum_{l=1}^{34} 1 = 136$ units. Now assume that with fused time steps, each layer $l$ can be processed in

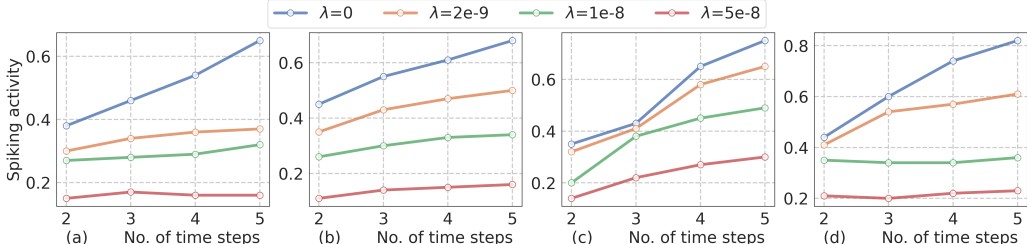

Figure 7: Comparison of the spiking activity of the SNNs obtained via our conversion method for different values of the regularization coefficient $\lambda$ for (a) VGG16 on CIFAR10, (b) ResNet20 on CIFAR10, (c) VGG16 on CIFAR100, and (d) ResNet20 on CIFAR100.

$t_{\text{layer}}(l) \approx 2$ units (accounting for parallel overhead less than 4). Then $D_{\text{layer}} \approx \sum_{l=1}^{34} 2 = 68$ units, half the latency of step-by-step. Even in a more conservative scenario where $t_{\text{layer}}(l)$ is, say, 3 units (not fully parallel), $D_{\text{layer}}$ would be 102 units, still $\sim 33\%$ faster than $D_{\text{step}}$. These numbers illustrate that throughput (in terms of latency per inference) is not sacrificed by layer-by-layer propagation; it can in fact improve it. Our method inherently exploits this by requiring only a small $T$, making fusion feasible. We also did not observe any degradation in empirical throughput on GPU during our experiments – the network with $T = 4$ ran efficiently, as all operations were implemented as matrix multiplies and thresholding that GPUs handle in batch.

In conclusion, the layer-by-layer propagation scheme is generally superior both in terms of spatial and latency complexity compared to the step-by-step propagation, and hence, our method that requires layer-by-layer propagation to operate successfully, does not incur any additional overhead.

### A.4 Latency & Energy Efficiency Details

Our proposed IF neuron model incurs the same addition, threshold comparison, and potential reset operations as that of a traditional IF model. It simply postpones the comparison and reset operations until after the input current is accumulated over all the $T$ time steps. Thus, our IF model has similar latency and energy complexity compared to the traditional IF model. Moreover, our proposed conversion framework requires that the output of each spiking convolutional layer is left-shifted by $(t-1)$ at the $t^{th}$ time step. However, as shown in Fig. 5, the number of left-shift operations in any network architecture is negligible compared to the total number of addition operations (even with the high sparsity provided by SNNs) incurred in the convolution operation. As a left-shift operation consumes similar energy as an addition operation for both 8-bit and 32-bit fixed point representation as shown in Table 6, the energy overhead of our proposed method is negligible compared to existing SNNs with identical spiking activity. Moreover, the energy overhead due to the addition, comparison and reset operation in our (this holds true for traditional IF models as well) IF model is also negligible compared to the spiking convolution operations as shown in Fig. 5.

Our SNNs yield high sparsity, thanks to our fine-grained $\ell_1$ regularizer, and ultra-low latency, thanks to our conversion framework. While the high sparsity reduces the compute energy compared to existing SNNs, the reduction compared to ANNs is significantly high. This is because ANNs incur multiplication operations in the convolutional layer which is $6.6-31\times$ more expensive compared to the addition operation as shown in Table 6. Thanks to the high sparsity ($71-79\%$) due to the $\ell_1$ regularizer, and the addition-only operations in our SNNs, we can obtain a $7.2-15.1\times$ reduction in the compute energy compared to an iso-architecture SNN, assuming a sparsity of $50\%$ due to the ANN ReLU layers.

The memory footprint of the SNNs during inference is primarily dominated by the read and write accesses of the post-synaptic potential at each time step (Datta et al., 2022; Yin et al., 2022). This is because these memory accesses are not influenced by the SNN sparsity since each post-synaptic potential is the sum of $k^2 c_{in}$ weight-modulated spikes. For a typical convolutional layer, $k = 3$, $c_{in} = 128$, and so it is almost impossible that all the $k^2 c_{in}$ spike values are zero for the membrane potential to be kept unchanged at a

| Model | Time Steps (T) | Latency per Sample (ms) | Accuracy (%) |
|---|---|---|---|
| Our Converted SNN | 4 | 8.2 | 95.82 |
| | 32 | 59.86 | 95.81 |
| QCFS-Converted SNN | 4 | 7.8 | 93.96 |
| | 32 | 59.63 | 95.54 |

Table 7: Comparison of latency and accuracy between our converted SNN and QCFS-converted SNN at different time steps, with the latency being profiled on NVIDIA Jetson Orin NX.

| Method | Network | QQP (%) | SST-2 (%) | QNLI (%) |
|---|---|---|---|---|
| QCSF | ANN | 84.04 | 81.44 | 80.92 |
| QCFS | SNN | 79.12 | 77.89 | 75.30 |
| **Ours** | SNN | 82.04 | 80.29 | 78.33 |

Table 8: Comparison of our proposed ANN-to-SNN conversion framework with QCFS for the BERT$_{BASE}$ network on a few representative GLUE tasks.

particular time step[2]. Since our proposed conversion framework significantly reduces the number of time steps compared to previous SNN training methods, it also reduces the number of membrane potential accesses proportionally. Hence, we reduce the memory footprint of the SNN during inference. However, it is hard to accurately quantify the memory savings since that depends on the system architecture and underlying hardware implementation.

Due to the exponentially lower number of time steps, our SNNs also incur lower latency compared to existing ANNs trained using ANN-to-SNN conversion. as shown in Table 7, our converted SNN model at T=4 achieves $\sim 7.3\times$ lower latency compared to the QCFS-converted SNN at T=32, while maintaining similar accuracy. Note that our neuron model maintains the same latency as the traditional IF model, as demonstrated by the comparable latency between our SNN and the QCFS-based SNN at identical time steps. These latency results are based on profiling inference per sample on the NVIDIA Jetson Orin GPU.

### A.5 Performance-Energy Trade-off with Bit-level regularizer

We can reduce the spiking activity of SNNs using our fine-grained $\ell_1$ regularizer. In particular, by increasing the value of the regularization co-efficient $\lambda$ from 0 to $5e-8$, the spiking activity can be reduced by $2.5-4.1\times$ for different architectures on CIFAR datasets as shown in Fig. 7. However, this comes at the cost of test accuracy, particularly for a very low number of time steps, $T<=3$, as shown in Fig. 6. By carefully tuning the value of $\lambda$, we can obtain SNN models with different sparsity-accuracy trade-offs that can be deployed in scenarios with diverse resource budgets. Using $\lambda=1e-8$ for the CIFAR datasets, and $\lambda=5e-10$ for ImageNet, yields a good trade-off for different time steps. As shown in Fig. 6, $\lambda=1e-8$ yields accuracies that are similar to $\lambda=0$. Note that $\lambda=0$ implies training of the source ANN without our fine-grained regularizer for $T\approx log_2 Q$ for CIFAR datasets. In particular, with ResNet18 for CIFAR10, $\lambda=1e-8$ yields SNN test accuracies within 0.2% of that of $\lambda=0$, while reducing the spiking activity by $\sim 2.4\times$ (0.53 to 0.22), which also reduces the compute energy by a similar factor. With ResNet34 for ImageNet, $\lambda=5e-10$, leads to a 0.4% reduction in test accuracy, while reducing the compute energy by $2\times$. Moreover, as shown in Fig. 7, the spiking activities of our SNNs trained with non-zero values of $\lambda$ do not increase significantly with the number of time steps as that with $\lambda=0$, which also demonstrates the improved compute efficiency resulting from our regularizer.

### A.6 Evaluation of Proposed Framework for Transformer Models

We also evaluate our ANN-to-SNN conversion framework on the BERT$_{BASE}$ model as shown in Table 8. We replace the GeLU activation function in the BERT model with the QCFS activation function to train the ANN, modified the SNN IF neuron model as proposed in our method. Note that, unlike CNNs, BERT

---

[2]Note that the number of weight read and write accesses can be reduced with the spike sparsity, and thus typically do not dominate the memory footprint of the SNN

| Architecture | Method | ANN | $T$=2 | $T$=4 | $T$=6 | $T$=8 | $T$=16 | $T$=32 |
|---|---|---|---|---|---|---|---|---|
| VGG16 | SNM | 74.13% | - | - | - | - | - | 71.80% |
| | SNNC-AP | 77.89% | - | - | - | - | - | 73.55% |
| | OPI | 76.31% | - | - | - | 60.49% | 70.72% | 74.82% |
| | BOS* | 76.28% | - | - | 76.03% | 76.26% | 76.62% | 76.92% |
| | QCFS | 76.28% | 63.79% | 69.62% | 72.50% | 73.96% | 76.24% | 77.01% |
| | **Ours** | 76.71% | 72.39% | 76.71% | 76.74% | 76.70% | 76.78% | 76.82% |
| ResNet20 | OPI | 70.43% | - | - | - | 23.09% | 52.34% | 67.18% |
| | BOS* | 69.97% | - | - | 64.21% | 65.18% | 68.77% | 70.12% |
| | QCFS | 69.94% | 19.96% | 34.14% | 49.20% | 55.37% | 67.33% | 69.82% |
| | **Ours** | 70.30% | 63.80% | 70.30% | 70.33% | 70.45% | 70.49% | 70.52% |

Table 9: Comparison of our proposed method to existing ANN-to-SNN Conversion approaches on CIFAR100 dataset. $Q$=16 for all architectures, and $\lambda$=$1e-8$. *BOS incurs 4 additional time steps to initialize the membrane potential, so the total number of time steps is $T$>4.

models do not have any batch normalization layer that succeeds the linear layer (unlike convolutional layer in CNNs), and hence, we could not eliminate the unevenness error by shifting any bias term. However, our modified neuron model outperforms the existing QCFS based conversion method by $\sim$2.8% on average for a range of tasks in the General Language Understanding Evaluation (GLUE) benchmark as shown below. We use $T = 16$ for a reasonable trade-off between accuracy and latency.

| Dataset | Approach | Architecture | Accuracy | Time steps |
|---|---|---|---|---|
| DSR | BPTT | ResNet18 | 73.35 | 4 |
| Diet-SNN | Hybrid | VGG16 | 69.67 | 5 |
| TEBN | BPTT | ResNet18 | 78.71 | 4 |
| IM-Loss | BPTT | VGG16 | 70.18 | 5 |
| RMP-Loss | BPTT | ResNet19 | 78.28 | 4 |
| SurrModu | BPTT | ResNet18 | 79.49 | 4 |
| **Our Work** | ANN-SNN | ResNet18 | **79.89** | 4 |

Table 10: Comparison of our proposed method with SOTA BPTT and hybrid training approaches on CIFAR100 dataset.

## A.7 Comparison with SOTA for CIFAR100

We compare our proposed framework with the SOTA ANN-to-SNN conversion approaches on CIFAR100 in Table 9. Similar to CIFAR10 and ImageNet, for ultra-low number of time steps, especially $T\leq4$, the test accuracy of our SNN models surpasses existing conversion methods. Moreover, our SNNs can also outperform SOTA-converted SNNs that incur even higher number of time steps. For example, the most recent conversion method, BOSQ reported a test accuracy of 76.03% at $T$=6 (with 4 time steps added on top of $T = 2$ in Table 9 for the extra 4 time steps required for potential initialization); our method can surpass that accuracy (76.71%) at $T$=4.

Additionally, as shown in Table 10, our ultra-low-latency accuracies are also higher compared to direct SNN training techniques, including BPTT and hybrid training step at iso-time-step. For example, our method can surpass the test accuracies obtained by the latest BPTT-based SNN training methods (Guo et al., 2023a; Deng et al., 2023) by 0.4$-$1.6%, while significantly reducing the training complexity.

| Dataset | Architecture | Neuromorphic | QANN | Bit-Serial |
|---------|--------------|--------------|------|------------|
| CIFAR10 | VGG16 | $1\times$ | $4.98\times$ | $3.57\times$ |
|         | ResNet18 | $1\times$ | $5.70\times$ | $4.54\times$ |
| ImageNet | VGG16 | $1\times$ | $4.52\times$ | $3.12\times$ |
|          | ResNet34 | $1\times$ | $5.12\times$ | $3.70\times$ |

Table 11: Comparison of normalized estimated energy of our SNNs on neuromorphic hardware compared quantized ANNs (QANN) and bit-serial ANNs.

### A.8 Comparison with Bit-Serial Quantized ANN

Bit-serial quantization is a popular implementation technique for neural network acceleration. It is often desirable for low precision hardware, including in-memory computing chips based on one-bit memory cells such as static random access memory (SRAM) and low-bit cells, such as resistive random access memory (RRAM). Similar to the SNN, it also requires a state variable that stores the intermediate bit-level computations, however, unlike the SNN that compares the membrane state with a threshold at each time step, it performs the non-linear activation function and produces the multi-bit output directly. However, to the best of our knowledge, there is no large-scale bit-serial accelerator chip currently available. Moreover, unlike neuromorphic chips, bit-serial accelerators do not leverage the large activation sparsity demonstrated in our work, and hence, incur significantly higher compute energy compared to neuromorphic chips. Since our SNNs trained with our bit-level regularizer provides a sparsity of $68-78\%$ for different architectures and datasets, they incur $3.1-4.5\times$ lower energy when run on sparsity-aware neuromorphic chips, compared to bit-serial accelerators, as shown in Table 11. Note that the values in Table 11 are derived by summing compute and memory energy based on established SNN energy models, as shown in the Section A.8 of Datta et al. (2024). Specifically, we estimate the number of floating point operations (FLOPs) and memory accesses in the SNN and apply energy-per-operation values from previous ASIC studies.

It can be argued that our approach without our bit-level regularizer leads to results similar to bit-serial computations. However, naively applying bit-serial computing to SNNs with the left-shift approach proposed in this work, would lead to non-trivial accuracy degradations. This is because unlike quantized networks, SNNs can only output binary spikes based on the comparison of the membrane potential against the threshold. Our proposed conversion optimization (bias shift of the BN layers and modification of the IF model) mitigates this accuracy gap, and ensures the SNN computation is identical to the activation-quantized ANN computation. This leads to zero conversion error from the quantized ANNs, and our SNNs achieve identical accuracy with the SOTA quantized ANNs.

### A.9 Memory Bandwidth Reduction with Proposed Method

Maintaining membrane potentials across time steps significantly impacts memory bandwidth usage in Spiking Neural Networks (SNNs). Each neuron's membrane potential must be read and updated at every time step; thus, increasing the number of time steps proportionally increases memory accesses, leading to higher bandwidth consumption. Conversely, reducing the number of time steps directly lowers the frequency of these accesses, thereby decreasing memory bandwidth demands.

For instance, in an SNN with $T = 16$ time steps, each neuron's membrane potential is accessed and updated 16 times per inference. Reducing $T$ to 4 decreases the number of accesses by a factor of 4, effectively reducing memory bandwidth consumption by 75%. In a VGG16 network designed for ImageNet classification with approximately 6.3 million neurons, this reduction translates to a fourfold decrease in memory accesses, assuming computation per time step remains constant.

The overall impact on memory energy depends on the hardware dataflow, particularly how weights and activations are stored and accessed. In a *weight-stationary dataflow*, where weights remain on-chip and are reused across multiple inputs, memory energy is often dominated by membrane potential accesses, as they must be dynamically computed at each time step. Consequently, reducing time steps can yield memory

| Epochs | Architecture | Type | Accuracy |
|--------|--------------|------|----------|
| 300 | VGG16 | QCFS pre-training | 95.82% |
| 30 | VGG16 | ReLU pre-training + QCFS fine-tuning | 95.47% |
| 300 | ResNet20 | QCFS pre-training | 93.60% |
| 30 | ResNet20 | ReLU pre-training + QCFS fine-tuning | 93.51% |

Table 12: Comparison of ANN training between QCFS pre-training and ReLU pre-training followed by QCFS fine-tuning for ANN-to-SNN conversion.

energy savings approaching the same factor as the time step reduction, making it a critical optimization for energy-efficient SNN implementations.

## A.10 Dependence on Training ANNs using QCFS Activation

While our ANN-to-SNN conversion framework is based on the QCFS activation function, it cannot be directly applied to ANNs trained using the ReLU function. However, our experimental results demonstrate that we need to fine-tune the ANNs with the QCFS function for only a small number of epochs when they are pre-trained with the ReLU function. In particular, as shown in Table 12, for both VGG16 and ResNet20, we only need 30 epochs of fine-tuning with the QCFS function for ANNs pre-trained with the ReLU function to achieve the same accuracy as training with the QCFS function for 300 epochs (as done in our original experiments).

## A.11 Extension to Object Detection

Model and Dataset: We chose a single-stage object detector architecture similar to RetinaNet with a ResNet-34 backbone (pre-trained on ImageNet) and detection heads for bounding box regression and classification. We evaluated on the MS COCO dataset, a standard benchmark for detection. The baseline ANN (ResNet34 backbone with feature pyramid and prediction heads) achieves an mAP (mean average precision) of around 30.1% on COCO validation – this is expected for a ResNet34-based detector (which is less powerful than ResNet50-based models that reach higher mAP).

Conversion Process: We applied our conversion framework to this detector. The backbone's convolutional layers all have batch normalization, which we adjusted (shifting the BN biases as per our method). The detection heads (for classification of proposals and for regression of box coordinates) do not contain BN layers in our implementation, but they do use ReLU activations. We converted those ReLUs to spiking neurons as well, using our proposed integrate-and-fire model. For the classification logits (which go into a sigmoid or softmax for object presence), we let the spiking neurons accumulate over $T$ time steps and take the firing rate as the analog value. For the bounding box regression outputs, which are real-valued, one straightforward approach is to represent them in a fixed-point spike rate as well. In our conversion, we treated the regression outputs as separate channels that were small in magnitude and could be encoded with a few time steps of spiking (essentially a minor extension to handle regression – we discretized the output range into a few spike counts). Importantly, no retraining of the detector was done after conversion; we directly used the ANN's trained weights.

We set $T = 4$ time steps for the SNN detector, consistent with our low-latency focus. This means each layer of the backbone and heads processes 4 spike cycles. Our modifications (like postponed resets) ensure that this $T = 4$ captures the necessary precision (in effect, equivalent to a 4-bit activation quantization for the features). The converted SNN detector achieves an mAP of 28.7% on COCO, which is remarkably close to the ANN's 30.1%. The drop of about 1.4 percentage points is modest and can be attributed mostly to the slight quantization of activation values due to having only 4 time steps. Comparing to recent works on spiking neural network detectors, such as a spiking RetinaNet achieved via direct training (Kim et al., AAAI 2020) that got mAP in the 20–30% range on COCO, our converted model's mAP 29% is on par with the state-of-the-art SNN detectors reported in literature (Zhang et al., 2023), despite us not doing any additional

SNN-specific training tricks. This is a strong indication that conversion is a viable path to bring SNNs into complex tasks like detection with minimal performance loss.

| Model | Time Steps (T) | mAP on COCO Validation (%) |
|---|---|---|
| Baseline ANN (ResNet-34 RetinaNet) | - | 30.1 |
| QCFS ANN | - | 28.7 |
| Converted SNN | 2 | 27.5 |
| Converted SNN | 4 | 28.7 |
| Converted SNN | 6 | 29.3 |

Table 13: Object detection performance comparison between the baseline ANN and converted SNNs with varying time steps (T).

| Model | Time Steps (T) | Noise Accuracy (%) | Adv. Attack Accuracy (%) |
|---|---|---|---|
| QCFS ANN | - | 88.7 | 85.2 |
| Converted SNN | 2 | 86.0 | 83.5 |
| Converted SNN | 4 | 86.5 | 83.0 |
| Converted SNN | 6 | 87.0 | 82.8 |

Table 14: Robustness performance comparison between the baseline ANN and converted SNNs with varying time steps (T).

## A.12 Robustness Analysis of the Proposed SNN

We conducted a brief experiment to gauge how our SNN handles random noise perturbations on input images, in comparison to the source ANN. On the CIFAR-10 test set, we introduced additive Gaussian noise to the images (zero-mean, with a standard deviation equal to 10% of the image pixel range). As shown in Table 14, the baseline ANN's accuracy dropped from 95.8% to 88.7% under this noise. Our converted SNN (ResNet20, $T = 4$) saw a very similar drop, from 93.6% to about 86.5%. The gap between ANN and SNN performance remained roughly the same ( 2–3% difference) under noise. This suggests that our conversion does not degrade the inherent robustness of the model: the SNN reacts to noisy inputs just as the ANN does. In some cases, we observed the SNN was slightly more tolerant to mild noise — likely because the spike thresholding can filter out very small fluctuations (a tiny input change might not cause an extra spike if it doesn't push the membrane over threshold). However, if the noise is significant, both ANN and SNN will be affected proportionally.

The enhanced robustness may be attributed of our SNNs may be attributed to the temporal integration, spike discretization, and inherited characteristics from the source ANN. Temporal robustness arises as neurons integrate information over T time steps, allowing transient noise to be averaged out, similar to a voting mechanism. While the effect is limited for small T, it still provides resilience against fleeting input perturbations. Binary robustness stems from the all-or-none nature of spikes—minor input changes below the threshold do not affect firing, unlike continuous activations in ANNs, which respond proportionally to small variations. Adversarial robustness largely depends on the source ANN; if the ANN is adversarially trained, this robustness is preserved in the converted SNN (Ozdenizci and Legenstein, 2024). While conversion alone does not inherently improve resistance to adversarial attacks, SNN discontinuities and temporal dynamics can complicate gradient-based adversarial manipulations, making some attacks less effective.

## A.13 Comparison with Binary Networks

Binary and ternary networks often experience a significant drop in accuracy, especially in tasks like ImageNet classification, due to their reduced representation capacity. Recent studies (Zhang et al., 2022; Bethge et al., 2021; Zhijun Tu and Wang, 2022; Xu et al., 2021; Shi et al., 2022) have attempted optimizations to mitigate this accuracy loss, but they still fall short of full-precision networks, as demonstrated in Table 15. In contrast,

| Network | Type | Dataset | Accuracy | Normalized CE |
|---|---|---|---|---|
| (Sakr et al., 2018) | BNN | CIFAR10 | 89.6 | $1\times$ |
| (Diffenderfer and Kailkhura, 2021) | BNN | CIFAR10 | 91.9 | $1\times$ |
| Ours | SNN | CIFAR10 | 95.82 | $1.15\times$ |
| (Zhang et al., 2022) | BNN | ImageNet | 73.4 | $1\times$ |
| (Bethge et al., 2021) | BNN | ImageNet | 71.0 | $1\times$ |
| Ours | SNN | ImageNet | 75.12 | $1.18\times$ |

Table 15: Comparison of our proposed ANN-to-SNN conversion method with T=4 against binary quantization approaches

| Architecture | T | Version | Accuracy (%) |
|---|---|---|---|
| VGG16 | 2 | PyTorch | 94.21% |
| | | Lava-DL | 94.15% |
| | 4 | PyTorch | 95.82% |
| | | Lava-DL | 95.61% |
| ResNet18 | 2 | PyTorch | 96.12% |
| | | Lava-DL | 95.77% |
| | 4 | PyTorch | 96.68% |
| | | Lava-DL | 96.02% |

Table 16: Comparison of the accuracies of our SNN models in PyTorch and Lava-DL.

our SNNs with T=4, even achieving a spike sparsity of 80%, consistently exceed the accuracies achieved by state-of-the-art binary networks.

Binary Neural Networks (BNNs) replace costly Multiply-Accumulate (MAC) operations with more affordable pop-count operations, leveraging binary weights and activations. While this reduces energy consumption compared to SNNs with multiple time steps, which involve additional operations per time step, the substantial spike sparsity enabled by our $\ell_1$ regularizer further mitigates this energy usage. Additionally, many state-of-the-art BNNs aim to enhance their expressive capacity through network modifications (e.g., ReactNet with PReLU), which significantly increase Floating Point Operations (FLOPs) by more than 2x compared to our simpler VGG and ResNets architectures. This trade-off between the combined factors of multiple time steps and additional operations versus pop-count operations and high spike sparsity in SNNs positions our SNN with T=4 to consume only 15-18% more compute energy than state-of-the-art BNNs, as depicted in Table 15. The compute energy is estimated from the energy model developed in Datta et al. (2024).

### A.14 Deployment of Proposed SNN on Loihi

Table 16 below compares the accuracies of our SNN in PyTorch and Lava-DL. We observed an average accuracy drop of approximately 0.3% on CIFAR-10 across both VGG and ResNet architectures when using the Lava-DL implementation compared to the PyTorch version. This discrepancy is likely due to the quantization of the weights and synaptic inputs inherent to the Lava-DL framework, which introduces slight computational differences. These results are included in the revised manuscript to provide a detailed analysis of the impact of deploying the SNN model on Loihi via Lava-DL.

### A.15 Pseudo code of proposed conversion framework

In this section, we summarize our proposed ANN-to-SNN conversion framework in Algorithm 1, which includes training the source ANNs using the QCFS activation function and then converting to SNNs.

---

**Algorithm 1** : Proposed ANN-to-SNN conversion algorithm

---

1: *Inputs*: ANN model $f^{ANN}(a; W, \mu, \sigma, \beta, \gamma)$ with initial weight $W$, BN layer running mean $\mu$, running variance $\sigma$, learnable scale $\gamma$, and learnable variance $\beta$; Dataset $D$; Quantization step $L$; Initial dynamic thresholds $\lambda$; Learning rate $\epsilon$; Number of SNN time steps $T$

2: *Output*: SNN model $f^{SNN}(a; W, \mu, \sigma, \beta, \gamma)$ & output $s^L(t) \; \forall t \in [1, T]$ where $L = f^{SNN}.\text{layers}$

3: #Source ANN training

4: **for** $e = 1$ to epochs **do**

5:     **for** length of dataset D **do**

6:         Sample minibatch $(a^0, y)$ from D

7:         **for** $l = 1$ to $f^{ANN}.\text{layers}$ **do**

8:             $a^l = \text{QCFS}(\gamma^l \left( \frac{W^l a^{l-1} - \mu^l}{\sigma^l} \right) + \beta^l)$

9:             $a^{i,l}_t = t^{th}$-bit, starting from MSB, of the $i^{th}$ term in $a^l$

10:         **end for**

11:         loss $= \text{CrossEntropy}(a^l, y) + \lambda \sum_{l=1}^{L} \sum_{t=1}^{T} a^{i,l}_t$

12:         **for** $l = 1$ to $f^{ANN}.\text{layers}$ **do**

13:             $W^l \leftarrow W^l - \epsilon \frac{\partial \text{loss}}{\partial W^l}, \;\; \mu^l \leftarrow \mu^l - \epsilon \frac{\partial \text{loss}}{\partial \mu^l}, \;\; \mu^l \leftarrow \sigma^l - \epsilon \frac{\partial \text{loss}}{\partial \sigma^l}$

14:             $\gamma^l \leftarrow \gamma^l - \epsilon \frac{\partial \text{loss}}{\partial \gamma^l}, \;\; \beta^l \leftarrow \beta^l - \epsilon \frac{\partial \text{loss}}{\partial \beta^l}, \;\; \lambda^l \leftarrow \lambda^l - \epsilon \frac{\partial \text{loss}}{\partial \lambda^l}$

15:         **end for**

16:     **end for**

17: **end for**

18: #ANN-to-SNN conversion

19: **for** $l = 1$ to $f^{ANN}.\text{layers}$ **do**

20:     $f^{SNN}.W^l \leftarrow f^{ANN}.W^l, f^{SNN}.\theta^l \leftarrow f^{ANN}.\lambda^l, \; f^{SNN}.\mu^l \leftarrow f^{ANN}.\mu^l, f^{SNN}.\sigma^l \leftarrow f^{ANN}.\sigma^l$

21:     $f^{SNN}.\gamma^l \leftarrow f^{ANN}.\gamma^l, f^{SNN}.\beta^l \leftarrow \frac{f^{ANN}.\beta^l}{T} + (1 - \frac{1}{T}) \frac{f^{ANN}.\gamma^l f^{ANN}.\mu^l}{f^{ANN}.\beta^l}$

22: **end for**

23: #Perform SNN inference on input $a^0$

24: $a^1 = \text{QCFS} \left( f^{SNN}.\gamma^1 \left( \frac{x^0 f^{SNN}.W^1 a^0 - f^{SNN}.\mu^1}{f^{SNN}.\sigma^1} \right) + f^{SNN}.\beta^1 \right)$

25: $s^1(t) = t^{th}$-bit of $a^1$ starting from MSB

26: **for** $l = 2$ to $f^{SNN}.\text{layers}$ **do**

27:     **for** $t = 1$ to $T$ **do**

28:         $z^l(t) = \left( f^{SNN}.\gamma^l \left( \frac{2^{t-1} f^{SNN}.W^l s^{l-1}(t) - f^{SNN}.\mu^l}{f^{SNN}.\sigma^l} \right) + f^{SNN}.\beta^l \right)$

29:     **end for**

30:     $u^l(1) = \sum_{t=1}^{T} z^l(t)$

31:     **for** $t = 1$ to $T$ **do**

32:         $s^l(t) = H(u^l(t) - \frac{f^{SNN}.\theta^l}{2})$

33:         $u^l(t+1) = u^l(t) - s^l(t) \frac{f^{SNN}.\theta^l}{2}$

34:     **end for**

35: **end for**

---

