# OpenReview forum: "When SNN meets ANN: Error-Free ANN-to-SNN Conversion for Extreme Edge Efficiency"
_TMLR — Accepted by TMLR_

### Review · Reviewer_67NH · 2025-02-13

**Summary Of Contributions:**

This paper presents a ANN-to-SNN conversion framework that achieves error-free conversion with significantly reduced time steps compared to existing approaches. The key contributions include a modified IF neuron model that postpones spike firing until after input accumulation, a batch normalization bias shift technique, and a fine-grained L1 regularizer to encourage spike sparsity. The authors demonstrate state-of-the-art accuracy on image classification tasks while requiring only 2-4 time steps, representing a major efficiency improvement over prior work requiring hundreds of time steps.

**Audience:**

Yes

**Broader Impact Concerns:**

N/A.

**Claims And Evidence:**

Yes

**Requested Changes:**

See my weakness part.

**Strengths And Weaknesses:**

### Strengths
- The theoretical foundation is rigorous, with detailed mathematical proofs showing how the proposed modifications eliminate conversion errors. The authors formally prove that their approach achieves exact matching between ANN and SNN activations under certain conditions.
- The experimental validation is comprehensive, covering multiple network architectures (VGG16, ResNet18/20/34, MobileNetV2) and datasets (CIFAR-10/100, ImageNet). The results consistently demonstrate superior accuracy-efficiency trade-offs compared to both ANN-to-SNN conversion and direct SNN training methods.
- The proposed L1 regularization technique for encouraging spike sparsity is novel and effective. The authors show 2.5-4.1x reduction in spike activity while maintaining competitive accuracy, which directly translates to energy savings on neuromorphic hardware.

### Weaknesses
- The layer-by-layer propagation requirement of the proposed method could potentially limit parallel processing capabilities. While the authors argue this has minimal impact on modern hardware, a more detailed analysis of the throughput implications would be valuable.
- The comparison with quantized ANNs could be more extensive. While the authors show energy advantages over 4-bit quantized networks, comparisons with more recent extreme quantization techniques (binary/ternary networks) are missing.
- The memory overhead analysis is somewhat limited. While spike sparsity reduces compute, the impact of maintaining membrane potentials across time steps on memory bandwidth isn't thoroughly quantified.
- The robustness of the method to different initialization schemes and training hyperparameters isn't extensively explored. Additional experiments showing sensitivity to these factors would strengthen the paper.
- The current approach requires retraining or fine-tuning ANNs with the QCFS activation function. While the authors show this can be done efficiently, direct conversion of pre-trained ReLU networks would be more practical for real-world adoption.
- The analysis of conversion errors focuses mainly on image classification tasks. Evaluation on other domains like object detection or semantic segmentation would better demonstrate the method's generalizability.

---

### Review · Reviewer_7Cxz · 2025-02-20

**Summary Of Contributions:**

The authors show that quantized artificial neural networks have an equivalent description as a discrete-time multi-layer Perceptron, which is characterized as a "spiking neural network". It is suggested that the resulting discrete-time system offers advantages in terms of latency and energy efficiency. A regularization term to encourage sparsity by reducing the number of active bits is proposed and shown to be effective. Authors provide extensive ablation studies and comparisons with existing work, demonstrating superior accuracy.

**Audience:**

Yes

**Broader Impact Concerns:**

None.

**Claims And Evidence:**

No

**Requested Changes:**

Requested changes to content:

* The manuscript claims "extreme energy efficiency" (6.5). However, it seems that the actual claim is that the proposed changes to the IF model to make it equivalent to the ANN do not incur a significant computation overhead relative to the original IF model. Authors should make it clear these claims regarding energy efficiency are made relative to other SNN models, and not on some absolute scale.

* Authors propose an exact mapping of quantized ANNs to SNNs. Under what circumstances does this provide a "free lunch" in terms of energy consumption? In other words, how can energy usage be reduced while the computation is fundamentally equivalent?

* Claiming "low latency" without further qualifiers suggests to the reader that the models can be used for faster inference based on wall-clock time. Authors should discuss what computational substrates for edge computing devices are required to actually achieve low(er) latency using lower numbers of timesteps in the models.

* Table 6: authors cite four sources as well as their own research ("in-house simulations"). Which numbers are obtained from which sources? The caption suggests that the numbers are based on a single 45nm ASIC.

* Table 10: How do you arrive at these numbers?

* Figure 5: What exactly is "spiking activity" and how is it determined for QCFS?

* Authors claim reductions in "compute energy", but never define this quantity. Authors need to provide a proper definition and cite sources using the same quantity.

* Authors should define the spaces that parameters and variables are part of (i.e., vectors, scalars etc)

* The proposed SNN bears no resemblance to biological models, but is formally equivalent to a discrete-time multi-layer Perceptron. However, what distinguishes SNNs from ANNs is precisely that spikes are discrete events in continuous time. Authors should reflect whether it is appropriate to call such a model an SNN, even if this use of the term is consistent with previous literature, and state explicitly what properties of their model set it apart from quantized ANNs or ANNs with binary activation functions.

Strongly suggested changes to form:

Authors should reconsider using the term "theorem" for mathematical statements with proofs that consist of rather trivial algebraic manipulations and substitutions. It seems more appropriate to use "proposition".

p. 7: Fig. 3, use different colors instead of different markers. The different markers are hard to differentiate visually.

p.8: Fig. 4, make the dashed lines much thicker. Right now it is hard to see them and differentiate from the grid lines.

p. 3, remove parentheses in citations: "In particular, (Guo et al., 2023a) and (Guo et al., 2022a)"

Suggested changes to form:
p.1: "However, since the binary spikes in the SNN layers need to be approximated with full-precision
ANN activations for accurate conversion, the number of SNN inference time steps required is high." -> "However, since the  full-precision
ANN activations need to be approximated by binary spikes in the SNN layers, the number of SNN inference time steps required is high."

p. 3: "Note that clip(x, 0, µ) = 0, if x < 0; x, if 0 ≤ x ≤ µ; µ, if x ≥ µ." Write this using {cases} environment for clarity

p. 6: "In contrast, (Liu et al., 2022)" -> lower case "in"

p. 17: When repeating equations, repeat the equations using the same number instead of a new number.

**Strengths And Weaknesses:**

While the article is generally well written, the strong and broad claims made by the authors ("extreme energy efficiency", "ultra-low latency") are insufficiently supported and the article requires significant revision before publication.

Strengths:
Generally well written. Authors provide extensive accuracy comparisons and ablation studies.

Weaknesses:
No convincing case is made for the strong claims in the abstract and conclusion. For example, it is apparently never stated how the authors arrive at the quantity "compute energy" they use to support their claims, or how the bottom-line energy for inference on a single sample would reduce based on the proposed conversion. Reductions in latency are only shown in a narrow sense: the number of time steps in the model can be reduced while maintaining accuracy. It could be argued that the original ANN has then the lowest possible latency (one time step). Authors never discuss how these "ultra-low latency" SNNs could translate into lower wall-clock latency, and what computational substrates would be required. It is unclear how the authors arrive at some numbers (e.g., table 6 and 10). Formatting issues like superfluous parentheses around citations (see requested changes) suggest carelessness in preparing the manuscript.

---

### Review · Reviewer_vBf2 · 2025-02-26

**Summary Of Contributions:**

The paper proposes a novel approach to converting Artificial Neural Networks (ANNs) to Spiking Neural Networks (SNNs), addressing the high time steps and spiking activity typically associated with conversion techniques. The framework modifies the standard integrate-and-fire (IF) neuron model in SNNs without altering the computational complexity. The proposed method produces lossless SNNs with low latency and energy consumption due to fewer time steps and high spike sparsity, while maintaining high test accuracy.

**Audience:**

Yes

**Claims And Evidence:**

Yes

**Requested Changes:**

See Weaknesses.

**Strengths And Weaknesses:**

Strengths
1. The writing is great, especially the theoretical analysis of conversion errors (Section 4).

2. The experiment results are remarkable, achieving 75.12% on ImageNet with only four time steps.

3. It is a novel method to conduct the bias shift within batch normalization layers.

Weaknesses
1. The results for ResNet-34 trained with the proposed method in Table 2 seem unusual. Specifically, why is the performance for T=6 significantly lower than for the other conditions? Theoretically, a larger number of time steps should always lead to better performance. Could you provide an analysis of this phenomenon? Additionally, it would be helpful to include standard deviations alongside the results for clarity.

2. Is the proposed conversion method applicable to training spiking Transformers, such as Spikformer [1] and Spike-driven Transformers [2][3]? Please discuss the potential for extending your method to larger SNN architectures.

[1] Zhou Z, Zhu Y, He C, et al. Spikformer: When Spiking Neural Network Meets Transformer[C]//The Eleventh International Conference on Learning Representations, 2023.

[2] Yao M, Hu J, Zhou Z, et al. Spike-driven transformer[C]. Advances in neural information processing systems, 2023, 36.

[3] Yao M, Hu J K, Hu T, et al. Spike-driven Transformer V2: Meta Spiking Neural Network Architecture Inspiring the Design of Next-generation Neuromorphic Chips[C]//The Twelfth International Conference on Learning Representations, 2024.

---

### Decision · Action_Editor_puJ3 · 2025-04-06

**Recommendation:** Accept with minor revision

**Comment:**

The paper proposes a novel approach to converting artificial neural networks (ANNs) into spiking neural networks (SNNs). It demonstrates that quantized artificial neural networks can be described equivalently as discrete-time multilayer perceptrons, which can, in turn, be characterized as spiking neural networks. The resulting discrete-time system offers advantages in terms of latency and energy efficiency. The mathematical proofs related to ANN-to-SNN conversion errors presented in the paper are intriguing, and the method shows high classification performance with only a few time steps, underscoring its utility. Although the initial version of the paper lacked sufficient support for the claims of “extremely energy efficiency” and “ultra-low latency,” these concerns have been addressed through the authors’ responses and revisions. Ultimately, the reviewers decided to accept the paper, and the AE agrees with the reviewers. There are some minor comments regarding the formatting of the references, which should be corrected in the final version.

**Audience:**

Spiking neural networks represent a somewhat niche area within the broader field of machine learning. On the other hand, addressing the various computational costs associated with the training and inference of large-scale deep neural networks, such as those used in large language models (LLMs), is a pressing issue for the TMLR community and constitutes an important topic. Furthermore, this paper provides a detailed mathematical proof demonstrating how conversion errors from artificial neural networks (ANNs) to spiking neural networks (SNNs) can be eliminated. The theoretical analysis process is likely to interest TMLR readers.

**Claims And Evidence:**

This paper proposes a novel approach for converting artificial neural networks (ANNs) into spiking neural networks (SNNs). In the paper, the authors claim “extremely energy efficiency” and “ultra-low latency” for their proposed method; however, the initial submission did not provide sufficient evidence to support these claims, which became a central point of discussion. The authors carefully addressed these concerns, conducted additional experiments, and revised the paper to include further explanations supporting their claims. As these responses and revisions adequately resolved the reviewers’ concerns, the claims in the paper are now considered to be well-supported by clear evidence.

---

> ### Author Response · Authors · 2025-04-29
> **Camera Ready Submission Uploaded**
>
> Dear Action Editor puJ3,
>
> We would like to sincerely thank you and the reviewers for the detailed and constructive feedback on our manuscript. The comments greatly helped us improve the quality and clarity of our work.
>
> We have carefully incorporated all the reviewer suggestions in the revised version. We are pleased to inform you that the camera-ready submission has now been uploaded.
>
> Thank you again for your time and support throughout the review process.
>
> Best,
>
> Authors

---

> > ### Comment · Reviewer_vBf2 · 2025-04-29
> > **About the references**
> >
> > Thanks for the authors' response, but I noticed that in the latest version, the names of the authors in the references I mentioned last time are still wrong (Page 16). Please check again.
> >
> > @inproceedings{ zhou2023spikformer, title={Spikformer: When Spiking Neural Network Meets Transformer }, author={Zhaokun Zhou and Yuesheng Zhu and Chao He and Yaowei Wang and Shuicheng YAN and Yonghong Tian and Li Yuan}, booktitle={The Eleventh International Conference on Learning Representations }, year={2023}, url={https://openreview.net/forum?id=frE4fUwz_h} }
> >
> > @inproceedings{yao2023spikedriven, title={Spike-driven Transformer}, author={Man Yao and JiaKui Hu and Zhaokun Zhou and Li Yuan and Yonghong Tian and Bo XU and Guoqi Li}, booktitle={Thirty-seventh Conference on Neural Information Processing Systems}, year={2023}, url={https://openreview.net/forum?id=9FmolyOHi5} }
> >
> > @inproceedings{ yao2024spikedriven, title={Spike-driven Transformer V2: Meta Spiking Neural Network Architecture Inspiring the Design of Next-generation Neuromorphic Chips}, author={Man Yao and JiaKui Hu and Tianxiang Hu and Yifan Xu and Zhaokun Zhou and Yonghong Tian and Bo XU and Guoqi Li}, booktitle={The Twelfth International Conference on Learning Representations}, year={2024}, url={https://openreview.net/forum?id=1SIBN5Xyw7} }

---

> > > ### Author Response · Authors · 2025-04-29
> > >
> > > Dear reviewer vBf2,
> > >
> > > Thank you for your latest comment. We have revised the submission accordingly by correcting the remaining reference errors and carefully reviewing the entire bibliography to ensure all entries are now accurate.